# Tachykinin acts upstream of autocrine Hedgehog signaling during nociceptive sensitization in *Drosophila*

**Seol Hee Im[1], Kendra Takle[2], Juyeon Jo[1,3], Daniel T Babcock[1,4†], Zhiguo Ma[2], Yang Xiang[2], Michael J Galko[1,3,4*]**

[1]Department of Genetics, University of Texas MD Anderson Cancer Center, Houston, United States; [2]Department of Neurobiology, University of Massachusetts Medical School, Worcester, United States; [3]Genes and Development Graduate Program, University of Texas Graduate School of Biomedical Sciences, Houston, United States; [4]Neuroscience Graduate Program, University of Texas Graduate School of Biomedical Sciences, Houston, United States

**\*For correspondence:** mjgalko@mdanderson.org

**Present address:** [†]Department of Genetics, University of Wisconsin-Madison, Madison, United States

**Competing interests:** The authors declare that no competing interests exist.

**Abstract** Pain signaling in vertebrates is modulated by neuropeptides like Substance P (SP). To determine whether such modulation is conserved and potentially uncover novel interactions between nociceptive signaling pathways we examined SP/Tachykinin signaling in a *Drosophila* model of tissue damage-induced nociceptive hypersensitivity. Tissue-specific knockdowns and genetic mutant analyses revealed that both *Tachykinin* and *Tachykinin-like receptor (DTKR99D)* are required for damage-induced thermal nociceptive sensitization. Electrophysiological recording showed that DTKR99D is required in nociceptive sensory neurons for temperature-dependent increases in firing frequency upon tissue damage. DTKR overexpression caused both behavioral and electrophysiological thermal nociceptive hypersensitivity. Hedgehog, another key regulator of nociceptive sensitization, was produced by nociceptive sensory neurons following tissue damage. Surprisingly, genetic epistasis analysis revealed that DTKR function was upstream of Hedgehog-dependent sensitization in nociceptive sensory neurons. Our results highlight a conserved role for Tachykinin signaling in regulating nociception and the power of *Drosophila* for genetic dissection of nociception.

## Introduction

Neuropeptides are key regulators of behavior. They can act as local neurotransmitters (*Salio et al., 2006*) or as tonic "gain controls" on neuronal activity to modify diverse aspects of organismal physiology including appetite, biological rhythms, aggression, and more (*Marder, 2012*; *Taghert and Nitabach, 2012*). Neuropeptide signaling also modulates nociception, the sensory perception of noxious stimuli. For example, Calcitonin Gene-Related Peptide (CGRP) and Substance P (SP) both regulate nociception in mammals (*Harrison and Geppetti, 2001*; *Seybold, 2009*). Modulation of nociception occurs following tissue damage, where the threshold that elicits aversive behaviors is reduced. This nociceptive sensitization can appear as allodynia - aversive responsiveness to previously innocuous stimuli, or hyperalgesia - exaggerated responsiveness to noxious stimuli (*Gold and Gebhart, 2010*). The exact roles of neuropeptides in regulating nociceptive sensitization are not yet clear.

In mammals, SP is highly expressed at the central nerve terminals of nociceptive sensory neurons where it is released as a peptide neurotransmitter (*Ribeiro-da-Silva and Hokfelt, 2000*). These neurons innervate the skin, are activated by noxious environmental stimuli, and project to second order

**eLife digest** Injured animals from humans to insects become extra sensitive to sensations such as touch and heat. This hypersensitivity is thought to protect areas of injury or inflammation while they heal, but it is not clear how it comes about.

Now, Im et al. have addressed this question by assessing pain in fruit flies after tissue damage. The experiments used ultraviolet radiation to essentially cause 'localized sunburn' to fruit fly larvae. Electrical impulses were then recorded from the larvae's pain-detecting neurons and the larvae were analyzed for behaviors that indicate pain responses (for example, rolling).

Im et al. found that tissue injury lowers the threshold at which temperature causes pain in fruit fly larvae. Further experiments using mutant flies that lacked genes involved in two signaling pathways showed that a signaling molecule called Tachykinin and its receptor (called DTKR) are needed to regulate the observed threshold lowering. When the genes for either of these proteins were deleted, the larvae no longer showed the pain hypersensitivity following an injury.

Further experiments then uncovered a genetic interaction between Tachykinin signaling and a second signaling pathway that also regulates pain sensitization (called Hedgehog signaling). Im et al. found that Tachykinin acts upstream of Hedgehog in the pain-detecting neurons. Following on from these findings, the biggest outstanding questions are: how, when and where does tissue damage lead to the release of Tachykinin to sensitize neurons? Future studies could also ask whether the genetic interactions between Hedgehog and Tachykinin (or related proteins) are conserved in other animals such as humans and mice.

neurons in laminae I of the spinal cord dorsal horn (*Allen et al., 1997*; *Marvizon et al., 1999*). These spinal neurons express a G-Protein-coupled receptor (GPCR), Neurokinin-1 receptor (NK-1R), which binds SP to transmit pain signals to the brain for further processing (*Brown et al., 1995*; *Mantyh et al., 1997*). NK-1R is also expressed in nociceptive sensory neurons (*Andoh et al., 1996*; *Li and Zhao, 1998*; *Segond von Banchet et al., 1999*). Once SP engages NK-1R, $G_q\alpha$ and $G_s\alpha$ signaling are activated leading to increases in intracellular $Ca^{2+}$ and cAMP (*Douglas and Leeman, 2011*). Whether other signal transduction pathways, especially other known mediators of nociceptive sensitization, are activated downstream of NK-1R is not known.

*Drosophila melanogaster* has several neuropeptides that are structurally related to SP. The *Drosophila Tachykinin (dTk)* gene encodes a prepro-Tachykinin that is processed into six mature Tachykinin peptides (DTKs) (*Siviter et al., 2000*). Two *Drosophila* GPCRs, TKR86C and TKR99D, share 32 – 48% identity to mammalian neurokinin receptors (*Li et al., 1991*; *Monnier et al., 1992*). All six DTKs and mammalian SP can activate TKR99D, increasing cytoplasmic $Ca^{2+}$ and cAMP levels (*Birse et al., 2006*). In *Drosophila, dTk* regulates gut contractions (*Siviter et al., 2000*), enteroendocrine homeostasis (*Amcheslavsky et al., 2014*; *Song et al., 2014*), stress resistance (*Kahsai et al., 2010a*; *Soderberg et al., 2011*), olfaction (*Ignell et al., 2009*), locomotion (*Kahsai et al., 2010b*), aggressive behaviors (*Asahina et al., 2014*), and pheromone detection in gustatory neurons (*Shankar et al., 2015*). Whether *dTk* and its receptors also regulate nociception and, if so, what downstream molecular mediators are involved have not yet been investigated.

*Drosophila* are useful for studying the genetic basis of nociception and nociceptive sensitization (*Im and Galko, 2011*). Noxious thermal and mechanical stimuli provoke an aversive withdrawal behavior in larvae: a 360-degree roll along their anterior-posterior body axis (*Babcock et al., 2009*; *Tracey et al., 2003*). This highly quantifiable behavior is distinct from normal locomotion and light touch responses (*Babcock et al., 2009*; *Tracey et al., 2003*). When a larva is challenged with a subthreshold temperature (38°C or below), only light touch behaviors occur, whereas higher thermal stimuli result in aversive rolling behavior (*Babcock et al., 2009*). Peripheral class IV multi-dendritic neurons (class IV neurons) are the nociceptive sensory neurons that innervate the larval barrier epidermis by tiling over it (*Gao et al., 1999*; *Grueber et al., 2003*) and mediate the perception of noxious stimuli (*Hwang et al., 2007*). For genetic manipulations within class IV neurons, *ppk1.9-GAL4* has been used widely as the 1.9 kb promoter fragment of pickpocket1 driving Gal4 selectively labels class IV nociceptive sensory neurons in the periphery (*Ainsley et al., 2003*). When the barrier epidermis is damaged by 254 nm UV light, larvae display both thermal allodynia and thermal hyperalgesia

(*Babcock et al., 2009*). This does not model sunburn because UV-C light does not penetrate the Earth's atmosphere, however, it has proven useful for dissecting the molecular genetics of nociceptive sensitization (*Im and Galko, 2011*).

What conserved factors are capable of sensitizing nociceptive sensory neurons in both flies and mammals? Known molecular mediators include but are not limited to cytokines, like TNF (*Babcock et al., 2009*; *Wheeler et al., 2014*), neuropeptides, metabolites, ions, and lipids (*Gold and Gebhart, 2010*; *Julius and Basbaum, 2001*). In addition, Hedgehog (Hh) signaling mediates nociceptive sensitization in *Drosophila* larvae (*Babcock et al., 2011*). Hh signaling regulates developmental proliferation and cancer (*Fietz et al., 1995*; *Goodrich et al., 1997*) and had not previously been suspected of regulating sensory physiology. The main signal-transducing component of the Hh pathway, *smoothened*, and its downstream signaling components, such as the transcriptional regulator *Cubitus interruptus* and a target gene *engrailed*, are required in class IV neurons for both thermal allodynia and hyperalgesia following UV irradiation (*Babcock et al., 2011*). In mammals, pharmacologically blocking Smoothened reverses the development of morphine analgesic tolerance in inflammatory or neuropathic pain models suggesting that the Smoothened/Hh pathway does regulate analgesia (*Babcock et al., 2011*). Interactions between the Hh and SP pathways in regulating nociception have not been investigated in either vertebrates or *Drosophila*.

Transient receptor potential (TRP) channels act as direct molecular sensors of noxious thermal and mechanical stimuli across phyla (*Venkatachalam and Montell, 2007*). In particular, the *Drosophila* TRPA family members, Painless (Pain) and TrpA1, mediate baseline thermal nociception in larvae (*Babcock et al., 2011*; *Tracey et al., 2003*; *Zhong et al., 2012*), as well as thermal sensation (*Kang et al., 2012*) and thermal nociception in adults (*Neely et al., 2010*). When larval class IV neurons are sensitized, it is presumably through modification of the expression, localization, or gating properties of TRP channels such as Painless or TrpA1. Indeed, direct genetic activation of either the TNF or Hh signaling pathway leads to thermal allodynia that is dependent on Painless. Direct genetic activation of Hh also leads to TrpA1-dependent thermal hyperalgesia (*Babcock et al., 2011*). Whether *Drosophila* TRP channels are modulated by neuropeptides like Tachykinin has not been addressed in the context of nociception.

In this study, we analyzed *Drosophila* Tachykinin and Tachykinin receptor (TkR99D or DTKR) in nociceptive sensitization. Both were required for UV-induced thermal allodynia: DTK from neurons likely within the central brain and DTKR within class IV peripheral neurons. Overexpression of DTKR in class IV neurons led to an ectopic hypersensitivity to subthreshold thermal stimuli that required specific downstream G protein signaling subunits. Electrophysiological analysis of class IV neurons revealed that when sensitized they display a DTKR-dependent increase in firing rates to allodynic temperatures. We also found that Tachykinin signaling acts upstream of *smoothened* in the regulation of thermal allodynia. Activation of DTKR resulted in a Dispatched-dependent production of Hh within class IV neurons. Further, this ligand was then required to relieve inhibition of Smoothened and lead to downstream engagement of Painless to mediate thermal allodynia. This study thus highlights an evolutionarily conserved modulatory function of Tachykinin signaling in regulating nociceptive sensitization, and uncovers a novel genetic interaction between Tachykinin and Hh pathways.

## Results

### Tachykinin is expressed in the brain and is required for thermal allodynia

To assess when and where Tachykinin might regulate nociception, we first examined DTK expression. We immunostained larval brains and peripheral neurons with anti-DTK6 (*Asahina et al., 2014*) and anti-*Leucopheae madurae* tachykinin-related peptide 1 (anti-LemTRP-1) (*Winther et al., 2003*). DTK was not detected in class IV neurons (*Figure 1—figure supplement 1*). Previous reports suggested that larval brain neurons express DTK (*Winther et al., 2003*). Indeed, numerous neuronal cell bodies in the larval brain expressed DTK and these extended tracts into the ventral nerve cord (VNC) (*Figure 1A*). Expression of a *UAS-dTk^{RNAi}* transgene via a pan-neuronal *Elav(c155)-GAL4* driver decreased DTK expression, except for a pair of large descending neuronal cell bodies in the protocerebrum (*Figure 1—figure supplement 2*) and their associated projections in the VNC, suggesting that these neurons express an antigen that cross-reacts with the anti-Tachykinin serum.

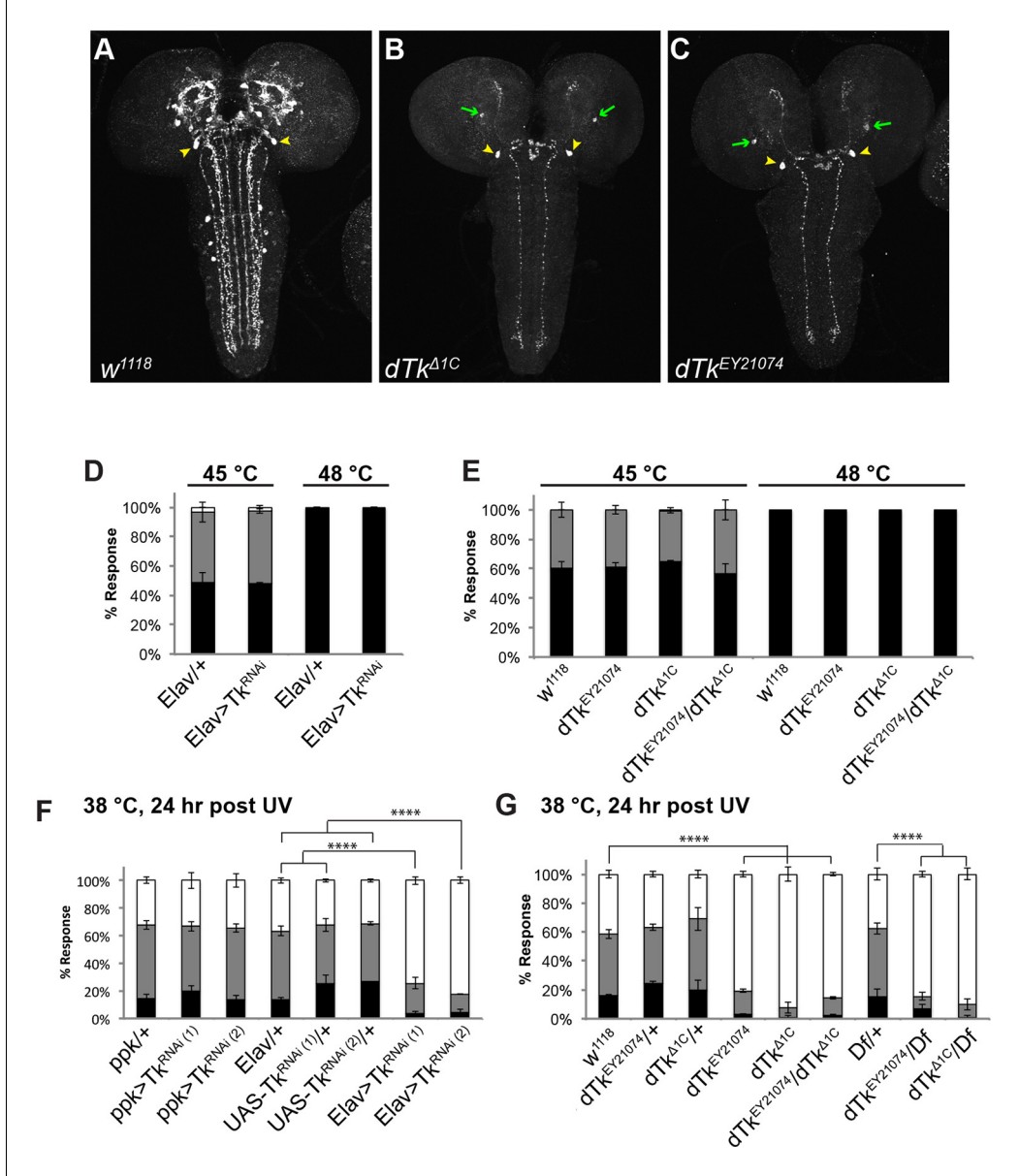

**Figure 1.** Tachykinin is expressed in the larval brain and required for thermal allodynia. (**A–C**) Dissected larval brain wholemounts of the indicated genotypes immunostained with a guinea pig antiserum to DTK6. Arrowheads, large immunoreactive descending neurons. Arrows, remaining neurons immunoreactive to anti-DTK6. (**A**) $w^{1118}$ (**B**) $dTk^{\Delta 1C}$ (**C**) $dTk^{EY21074}$ (**D**) Baseline responses to thermal stimulation in the absence of injury at 45°C and 48°C when Tachykinin is targeted by RNAi in all neurons. Larvae of indicated genotypes were stimulated for up to 20 s with a thermal probe set to the indicated temperatures. The resulting behavior was categorized as "no withdrawal" (white) if a 360 ° aversive roll did not occur, "slow withdrawal" (gray), if the roll occurred between 6 and 20 s of probe contact, or "fast withdrawal" (black), if the roll occurred within 5 s of probe contact. Percent behavioral responses were plotted as mean ± SEM. This scheme was employed for all behavioral quantitation in this study. (**E**) Baseline responses to thermal stimulation at 45°C and 48°C of $dTk$ mutant alleles and relevant controls. (**F–G**) UV-induced thermal allodynia. (**F**) RNAi targeting $dTk$ and controls. (1) and (2) refer to non-overlapping *UAS-RNAi* transgenes targeting Tachykinin. (**G**) Mutant alleles of $dTk$ and controls. All behavior experiments throughout were performed in triplicate sets of n = 30 unless noted otherwise. Statistical significance was determined by the chi-square test. Same statistical significance markers were used throughout all figures. *$p<0.05$, **$p<0.01$, ***$p<0.001$, ****$p<0.0001$.

The following figure supplements are available for figure 1:

*Figure 1 continued*

**Figure supplement 1.** Tachykinin is not expressed in class IV md nociceptive sensory neurons.
**Figure supplement 2.** Dissected larval brain whole mounts of *Elav/+* and *Elav>TK^RNAi* immunostained with anti-LemTRP.
**Figure supplement 3.** Schematic of the *dTk* locus.
**Figure supplement 4.** Temperature versus behavior dose response curves.
**Figure supplement 5.** Alternative data presentation of thermal allodynia (a subset of *Figure 1F* and a subset of *Figure 1G*) in non-categorical line graphs of accumulated percent response as a function of measured latency.

Labeling of anti-DTK6 in the brain was also greatly decreased (*Figures 1B and C*) in homozygous larvae bearing two different *dTk* alleles, *dTk^EY21074* and *dTk^Δ1C*, that decrease Tachykinin function (*Figure 1—figure supplement 3*). Therefore, we conclude that *dTk* expression is effectively knocked down both in mutants and by RNAi transgenes.

Because we observed a knockdown of DTK staining in the brain with mutants and RNAi, and because mammalian SP regulates pain behavior, we tested if *dTk* loss of function affects nociceptive behaviors. We first tested baseline nociception in the absence of injury, where larvae were challenged with noxious thermal stimuli at 45°C or 48°C, the middle and upper end of their response range, respectively (*Babcock et al., 2009*). For uninjured larvae, the behavioral dose-response to temperature forms a reproducible graded curve (*Figure 1—figure supplement 4*). Pan-neuronal knockdown of *dTk* did not cause baseline nociception defects compared to relevant *GAL4* controls (*Figure 1D*). Similarly, larvae homozygous or transheterozygous for *dTk^EY21074* or *dTk^Δ1C* had normal baseline thermal nociceptive responses (*Figure 1E*).

Next, we tested UV-induced nociceptive sensitization. Pan-neuronal knockdown of *dTk* significantly reduced thermal allodynia (responsiveness to sub-threshold 38°C) (*Figure 1F* and *Figure 1—figure supplement 5*). Two non-overlapping RNAi transgenes (*Tk^JF01818* and *Tk^KK112227*) targeting *Tachykinin* reduced the allodynia response from 70% to about 20% compared to relevant *GAL4* or *UAS* alone controls 24 hr after UV irradiation (*Figure 1F*). Consistent with the absence of DTK staining in class IV neurons (*Figure 1—figure supplement 1*), class IV-specific knockdown of *dTk* did not alter thermal allodynia (*Figure 1F*). As genetic confirmation of the RNAi phenotype, we tested mutant alleles of *dTk* for tissue damage-induced thermal allodynia. Heterozygous larvae bearing these *dTk* alleles, including a deficiency spanning the *dTk* locus, displayed normal thermal allodynia (*Figure 1G*). By contrast, all homozygous and transheterozygous combinations of *dTk* alleles drastically reduced thermal allodynia (*Figure 1G*). Therefore, we conclude that *Tachykinin* is necessary for the development of thermal allodynia in response to UV-induced tissue damage.

## Tachykinin Receptor is required in class IV nociceptive sensory neurons for thermal allodynia

Two GPCRs recognize Tachykinins. DTKR (*TkR99D* or *CG7887*) recognizes all six DTKs (*Birse et al., 2006*) whereas NKD (*TkR86C* or *CG6515*) binds DTK-6 and a tachykinin-related peptide, natalisin (*Jiang et al., 2013*; *Monnier et al., 1992*; *Poels et al., 2009*). Because DTKR binds more broadly to DTKs, we tested if class IV neuron-specific knockdown of *dtkr* using the *ppk-GAL4* driver (*Ainsley et al., 2003*) led to defects in either baseline nociception or thermal allodynia. See *Figure 2A* for a schematic of the *dtkr* locus and the genetic tools used to assess this gene's role in thermal allodynia. Similar to *dTk*, no baseline nociception defects were observed upon knockdown of *dtkr* (*Figure 2B*). Larvae homozygous for *TkR99D^f02797* and *TkR99D^MB09356* were also normal for baseline nociceptive behavior (*Figure 2C*).

Although baseline nociception was unaffected, class IV neuron-specific expression of *UAS-dtkr^RNAi* significantly reduced thermal allodynia compared to *GAL4* or *UAS* alone controls (*Figure 2D* and *Figure 2—figure supplement 1*). This reduction was rescued upon simultaneous overexpression of DTKR using a *UAS-DTKR-GFP* transgene, suggesting that the RNAi-mediated

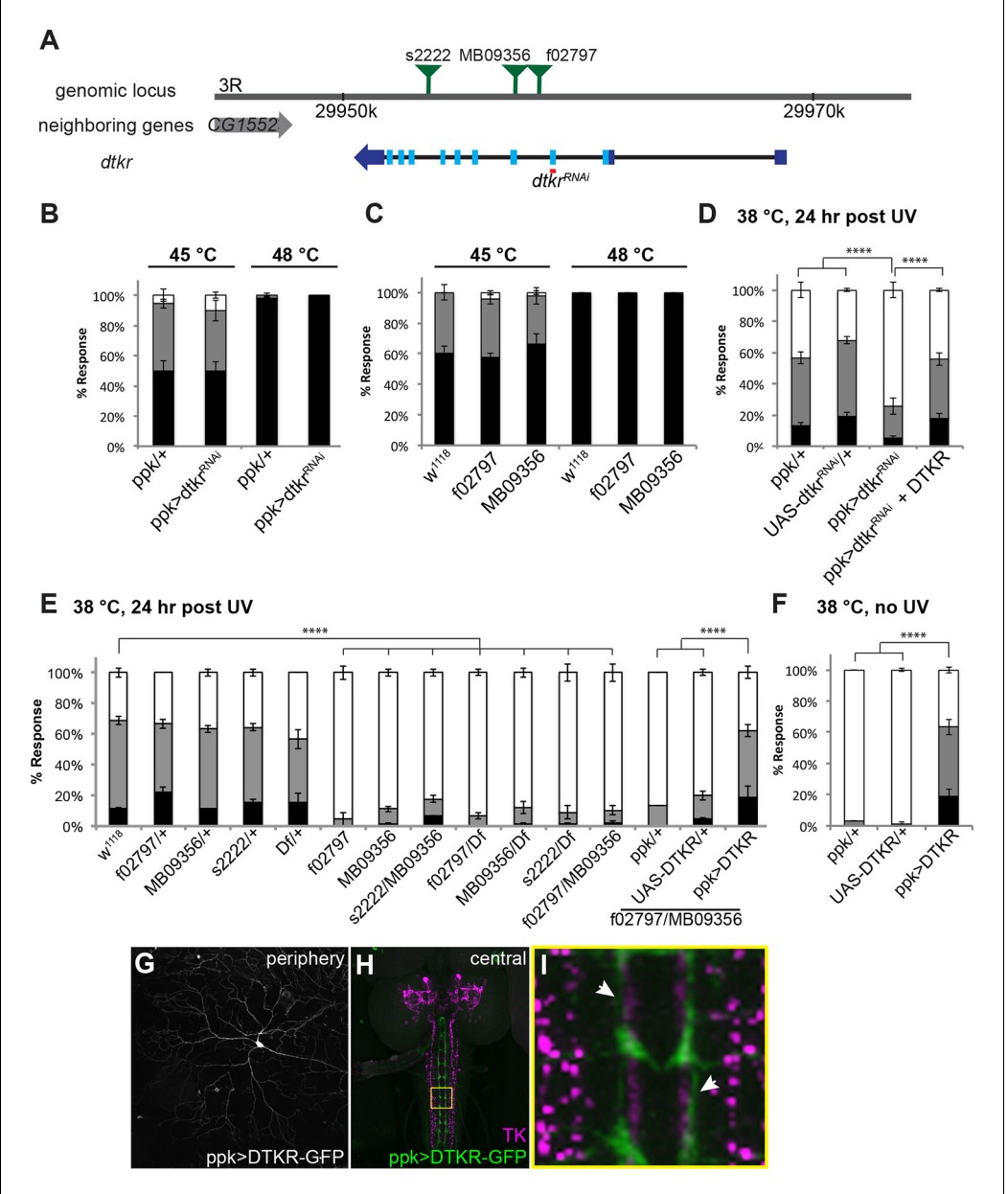

**Figure 2.** Tachykinin Receptor is required in class IV nociceptive sensory neurons for thermal allodynia. (**A**) Schematic of the *dtkr* genomic locus. Location of transposon insertion alleles and targeted sequences of *UAS-RNAi* transgenes are shown. (**B,C**) Baseline thermal nociception at 45°C and 48°C. (**B**) *dtkr* RNAi in class IV neurons and controls. (**C**) *dtkr* mutant alleles and controls. (**D,E**) UV-induced thermal allodynia at 38°C. (**D**) *dtkr* RNAi and rescue in class IV neurons. (**E**) *dtkr* mutant alleles and controls. (**F**) "Genetic" thermal allodynia in the absence of injury upon overexpression of DTKR in class IV neurons. (**G–I**) Dissected larval epidermal wholemounts (genotype: *ppk>DTKR-GFP)* immunostained with anti-LemTRP-1 (reacts to DTKs) and anti-GFP. (**G**) DTKR-GFP expression in class IV neuron soma and dendrites. (**H**) Larval brain wholemount. GFP (green); anti-DTK (magenta). Yellow Box indicates close-up shown in I. (**I**) Axonal tracts expressing DTKR-GFP in class IV neurons juxtaposed with TK-expressing cells in the VNC. Arrows, regions where GFP-expressing axons are closely aligned with DTK-expressing axons.

The following figure supplement is available for figure 2:

**Figure supplement 1.** Alternative data presentation of thermal allodynia (*Figure 2D* and a subset of *Figure 2E*) in non-categorical line graphs of accumulated percent response as a function of measured latency.

phenotype was not off-target (*Figure 2D*). We also tested mutant alleles of *dtkr* for thermal allodynia defects. While all heterozygotes were normal, larvae bearing any homozygous or transheterozygous combination of alleles, including a deficiency spanning the *dtkr* locus, displayed greatly reduced thermal allodynia (*Figure 2E*). Restoration of DTKR expression in class IV neurons in a *dtkr* mutant background fully rescued their allodynia defect (*Figure 2E* and *Figure 2—figure supplement 1*) suggesting that the gene functions in these cells. Lastly, we examined whether overexpression of DTKR within class IV neurons could ectopically sensitize larvae. While *GAL4* or *UAS* alone controls remained non-responsive to sub-threshold 38°C, larvae expressing DTKR-GFP within their class IV neurons showed aversive withdrawal to this temperature even in the absence of tissue damage (*Figure 2F*). Visualization of the class IV neurons expressing DTKR-GFP showed that the protein localized to both the neuronal soma and dendritic arbors (*Figure 2G*). Expression of DTKR-GFP was also detected in the VNC, where class IV axonal tracts run immediately adjacent to the axonal projections of the Tachykinin-expressing central neurons (*Figures 2H and I*). Taken together, we conclude that DTKR functions in class IV nociceptive sensory neurons to mediate thermal allodynia.

## Tachykinin signaling modulates firing rates of class IV nociceptive sensory neurons following UV-induced tissue damage

To determine if the behavioral changes in nociceptive sensitization reflect neurophysiological changes within class IV neurons, we monitored action potential firing rates within class IV neurons in UV- and mock-treated larvae. As in our behavioral assay, we UV-irradiated larvae and 24 hr later monitored changes in response to thermal stimuli. Here we measured firing rates with extracellular recording in a dissected larval fillet preparation (*Figure 3A* and methods). Mock-treated larvae showed no increase in their firing rates until around 39°C (*Figures 3B and D*). However, UV-treated larvae showed an increase in firing rate at temperatures from 31°C and higher (*Figures 3C and D*). The difference in change in firing rates between UV- and mock-treated larvae was significant between 30°C and 39°C. This increase in firing rate demonstrates sensitization in the primary nociceptive sensory neurons and correlates well with behavioral sensitization monitored previously.

Next, we wondered if loss of *dtkr* could block the UV-induced increase in firing rate. Indeed, class IV neurons of *dtkr* mutants showed little increase in firing rates even with UV irradiation (*Figure 3E*). Similarly, knockdown of *dtkr* within class IV neurons blocked the UV-induced increase in firing rate; UV- and mock-treated *UAS-dtkr^RNAi^*-expressing larvae showed no statistically significant difference in firing rate (*Figure 3E*). When DTKR expression was restored only in the class IV neurons in the *dtkr* mutant background, the firing rates increased with increasing temperature upon UV irradiation (*Figure 3E* and see *Figure 3—figure supplement 1* for additional control genotypes). Thus, *dtkr* functions in class IV neurons for the UV-induced increase in firing rate in response to increasing temperature.

Next we overexpressed DTKR in class IV neurons and tested the effect of gain of function on the neuronal firing rate. Behaviorally, overexpression induced ectopic sensitization even without UV (*Figure 2F*). When we assayed lower temperatures (32–38°C), the ectopic thermal allodynia was obvious above 34°C (*Figure 3H*). Electrophysiologically, we saw similar results. Class IV neurons expressing DTKR-GFP increased their firing rate to thermal stimuli even without UV irradiation (*Figures 3F–H*). The magnitude of the increase upon overexpression was comparable to that of UV-treated controls (*Figures 3D and H*). Taken together, electrophysiological recordings corresponded well with the behavioral changes seen upon loss- or gain-of-function of Tachykinin signaling. The electrophysiology further suggests that DTKR signaling modulates the firing properties of class IV nociceptive sensory neurons in response to tissue damaging stimuli like UV radiation.

## Trimeric G proteins act downstream of Tachykinin signaling in thermal allodynia

DTKR activation increases cytoplasmic $Ca^{2+}$ and cAMP levels in a heterologous cell-based assay (*Birse et al., 2006*), suggesting receptor coupling to $G_{\alpha}s$ and/or $G_{\alpha}q$. To identify the particular trimeric G-protein subunits through which Tachykinin and its receptor modulate thermal allodynia, we screened five of six annotated and two putative Gα-, all three annotated and one putative Gβ-, and all two annotated and one putative Gγ-encoding genes (*Figure 4A*). Several *UAS-RNAi* transgenes yielded modest defects in thermal allodynia (*Figure 4B* and *Figure 4—figure supplement 1*). When

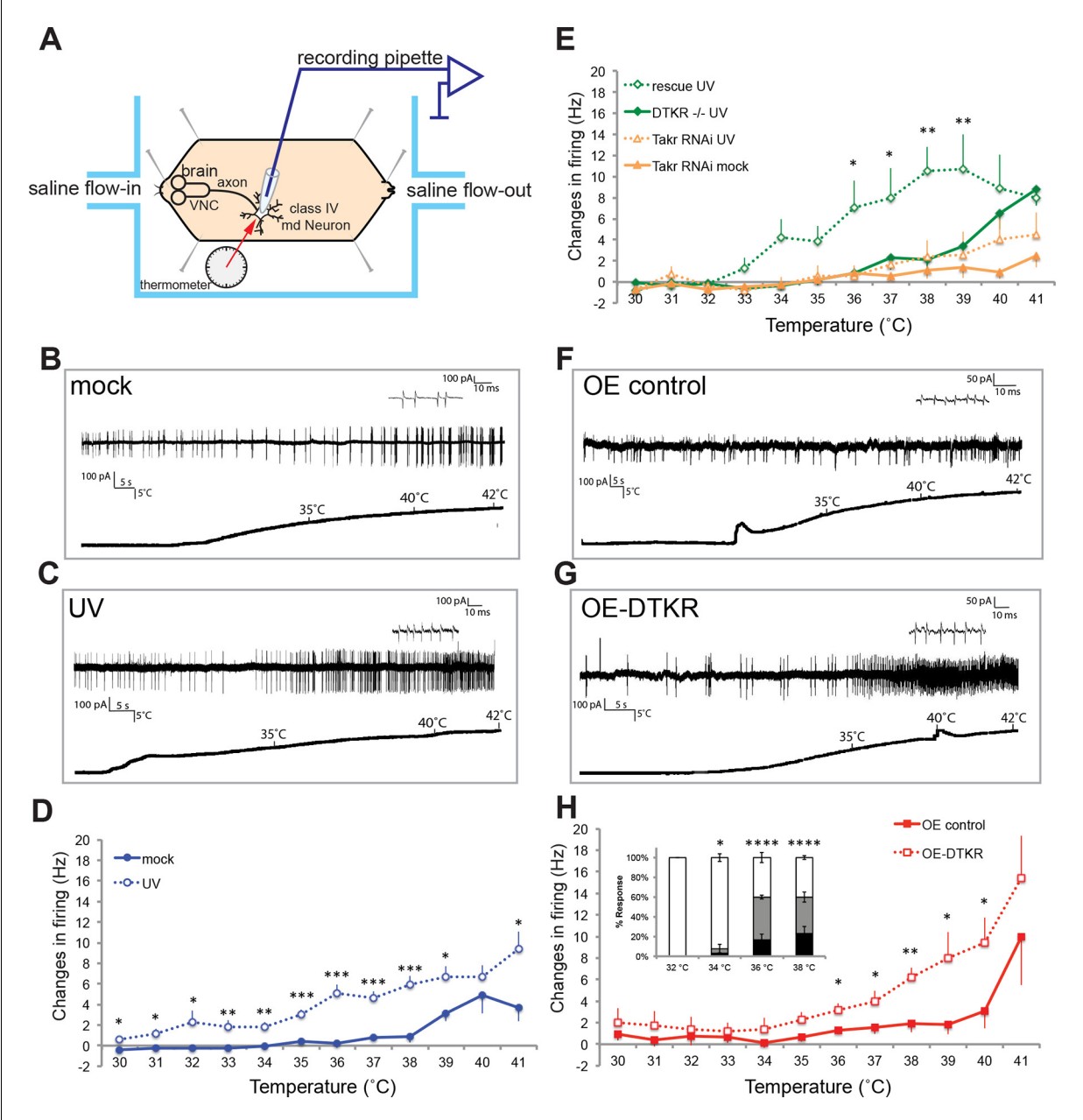

**Figure 3.** Class IV neurons display temperature-dependent changes in firing rates that are modulated by Tachykinin signaling. (**A**) Schematic diagram of assay setup. (**B,C,F,G**) Sample recording traces of the indicated genotypes in response to temperature ramping. (**B**) *ppk1.9-Gal4, ppk-eGFP/+* mock (**C**) *ppk1.9-Gal4, ppk-eGFP/+* 24 hr following UV (**F**) *ppk-Gal4/+* (**G**) *ppk-Gal4>DTKR-GFP*. (**D**) Changes in firing rates from larvae in (**B**) and (**C**) in response to temperature ramping. n = 11 (mock), and 21 (UV). (**E**) Changes in firing rates between *ppk-Gal4>dtkr^RNAi* (mock and UV), *dtkr^MB09356/f02797* (UV), and class IV neuron-specific rescue of *dtkr^MB09356/f02797* (UV) in response to temperature ramping. n = 12 (RNAi mock), 11 (RNAi UV), 17 (*dtkr* mutant), and 12 (rescue). (**H**) Changes in firing rates between Gal4 only control and class IV specific overexpression of DTKR in response to temperature ramping without tissue damage. n = 9 (control), 12 (Overexpression). Inset, Behavioral response to innocuous temperatures when DTKR is overexpressed in class IV neurons without tissue damage. *= P<0.05, **= P<0.01, ***= P<0.001. Statistical significance was determined by either Two-way ANOVA test with Bonferroni correction or two-tailed unequal variance Student's t-Test for electrophysiology, or by Chi-square analysis for behavior analysis.

The following figure supplement is available for figure 3:

**Figure supplement 1.** Control genotypes for electrophysiology recordings of class IV neurons.

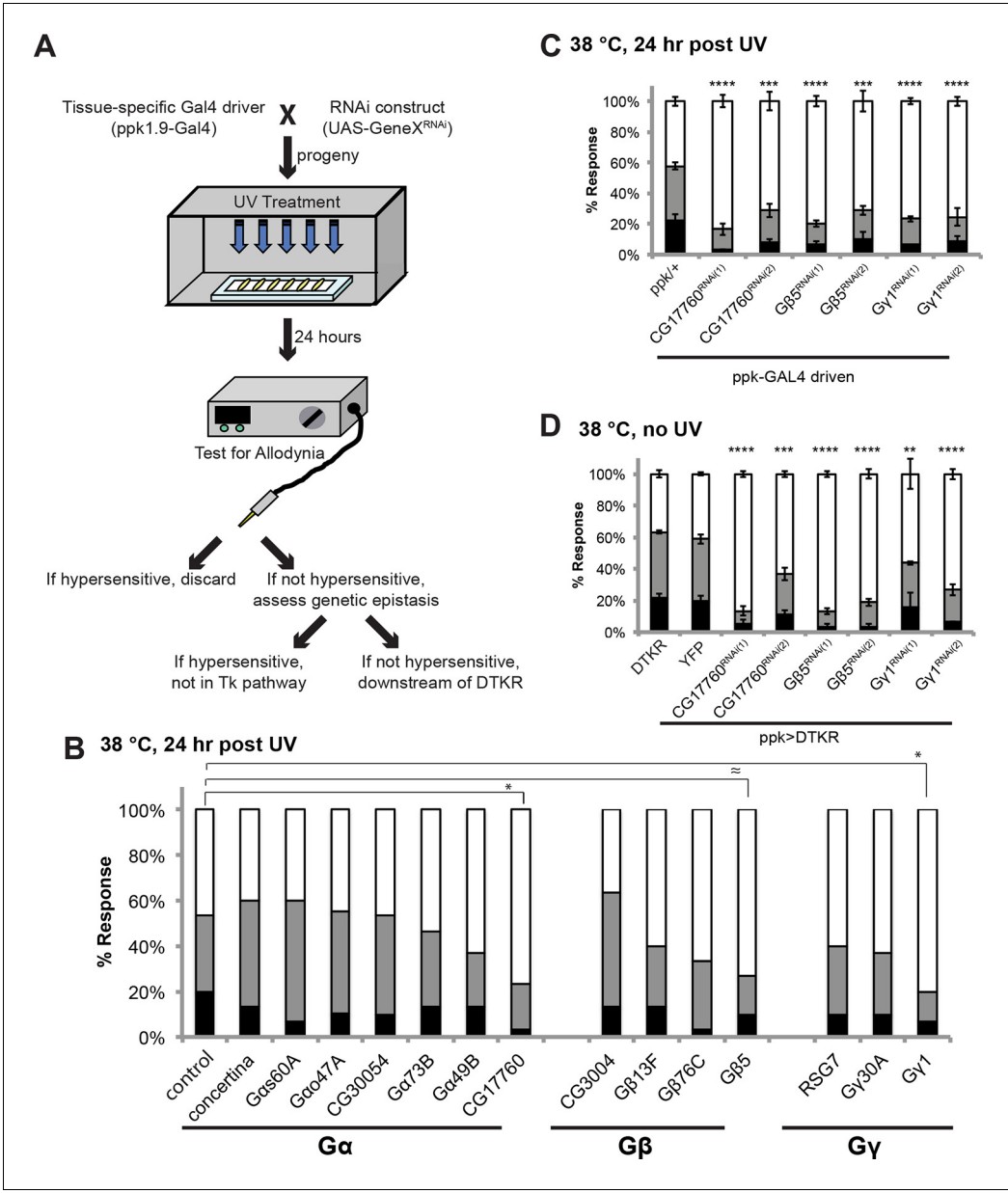

**Figure 4.** Specific Trimeric G proteins act downstream of DTKR in class IV neurons in thermal allodynia. (**A**) Schematic of genetic screening strategy for testing G-protein subunit function by in vivo tissue-specific RNAi in class IV neurons. (**B**) UV-induced thermal allodynia on targeting the indicated G protein subunits by RNAi. n = 30 larvae per genotype. ≈ P = 0.082, * P<0.05. Statistical significance was determined by Fisher's exact test. (**C**) UV-induced thermal allodynia for the three putative hits from the mini-screen in A. (1) and (2) indicate non-overlapping RNAi transgenes. (**D**) Suppression of *UAS-DTKR*-induced "genetic" allodynia by co-expression of *UAS-RNAi* transgenes targeting the indicated G protein subunits. Seven sets of n=30 for *ppk>DTKR-GFP* controls, triplicate sets of n=30 for the rest.

The following figure supplements are available for figure 4:

**Figure supplement 1.** Alternative data presentation of UV-induced thermal allodynia on targeting G protein subunits by RNAi (*Figure 4B*) in non-categorical line graphs of accumulated percent response as a function of measured latency.

**Figure supplement 2.** UAS alone controls of RNAi targeting G protein subunits do not exhibit defects in UV-induced thermal allodynia.

analyzing our behavioral data categorically, Gβ5 was not quite significant, but when the data was analyzed non-categorically (accumulated percent response versus latency) the increased statistical power of this method revealed that Gβ5 was significantly different from the control (*Figure 4—figure supplement 1*). Indeed, retesting the strongest hits in greater numbers and analyzing them categorically revealed that knockdown of a putative G$_{\alpha}$q (*CG17760*), *Gβ5* (CG10763), and Gγ1 (CG8261) all significantly reduced thermal allodynia compared to *GAL4* and UAS-alone controls (*Figure 4C* and *Figure 4—figure supplements 1* and *2*). To test if these subunits act downstream of DTKR, we asked whether expression of the relevant *UAS-RNAi* transgenes could also block the ectopic thermal allodynia induced by DTKR-GFP overexpression (*Figure 2F*). All of them did (*Figure 4D*). Therefore, we conclude that *CG17760, Gβ5*, and Gγ1 are the downstream G protein subunits that couple to DTKR to mediate thermal allodynia in class IV neurons.

## Tachykinin signaling acts upstream of Smoothened and Painless in allodynia

The signal transducer of the Hedgehog (Hh) pathway, Smoothened (*smo*), is required within class IV neurons for UV-induced thermal allodynia (*Babcock et al., 2011*). To determine if Tachykinin signaling genetically interacts with the Hh pathway during thermal allodynia, we tested the behavior of a double heterozygous combination of *dtkr* and *smo* alleles. Such larvae are defective in UV-induced thermal allodynia compared to relevant controls (*Figure 5A* and *Figure 5—figure supplement 1*).

We next performed genetic epistasis tests to determine whether Tachykinin signaling functions upstream, downstream, or parallel of Hh signaling during development of thermal allodynia. The general principle was to co-express an activating transgene of one pathway (which induces genetic thermal allodynia) together with an inactivating transgene of the other pathway. Reduced allodynia would indicate that the second pathway was acting downstream of the ectopically activated one (see schematic of possible outcomes in *Figure 5B*). For example, to test if Tachykinin signaling is downstream of *smo*, we combined a dominant negative form of Patched (*UAS-Ptc$^{DN}$*) that constitutively activates Smo and causes ectopic thermal allodynia (*Babcock et al., 2011*) with *UAS-dtkr$^{RNAi}$*. This did not block the ectopic sensitization (*Figure 5C*) while a positive control gene downstream of *smo* did (*UAS-engrailed$^{RNAi}$*), suggesting that *dtkr* does not function downstream of *smo*. In a converse experiment, we combined *UAS-DTKR-GFP* with a number of transgenes capable of interfering with Smo signal transduction. Inactivation of Smo signaling via expression of *Patched (UAS-Ptc)*, or a dominant negative form of *smo (UAS-smo$^{DN}$)*, or a dominant negative form of the transcriptional regulator *Cubitus interruptus (UAS-Ci$^{DN}$)*, or an RNAi transgene targeting the downstream transcriptional target *engrailed (UAS-en$^{RNAi}$)*, all abolished the ectopic sensitization induced by overexpression of DTKR-GFP (*Figure 5D* and *Figure 5—figure supplement 1*). Thus, functional Smo signaling components act downstream of DTKR in class IV neurons.

The TNF receptor Wengen (*Kanda et al., 2002*) is required in class IV nociceptive sensory neurons to elicit UV-induced thermal allodynia (*Babcock et al., 2009*). We therefore also tested the epistatic relationship between DTKR and the TNFR/Wengen signaling pathways and found that they function independently of/in parallel to each other during thermal allodynia (*Figure 5—figure supplement 2*). This is consistent with previous genetic epistasis analysis, which revealed that TNF and Hh signaling also function independently during thermal allodynia (*Babcock et al., 2011*).

The TRP channel *pain* is required for UV-induced thermal allodynia downstream of Smo (*Babcock et al., 2011*). Because Smo acts downstream of Tachykinin this suggests that *pain* would also function downstream of *dtkr*. We formally tested this by combining DTKR overexpression with two non-overlapping *UAS-pain$^{RNAi}$* transgenes. These *UAS-pain$^{RNAi}$* transgenes reduced baseline nociception responses to 48°C although not as severely as *pain$^{70}$*, a deletion allele of *painless* (*Figure 5—figure supplement 3,4* and . As expected, combining DTKR overexpression and *pain* knockdown or DTKR and *pain$^{70}$* reduced ectopic thermal allodynia (*Figure 5E*). In sum, our epistasis analysis indicates that the Smo signaling cassette acts downstream of DTKR in class IV neurons and that these factors then act via Painless to mediate thermal allodynia.

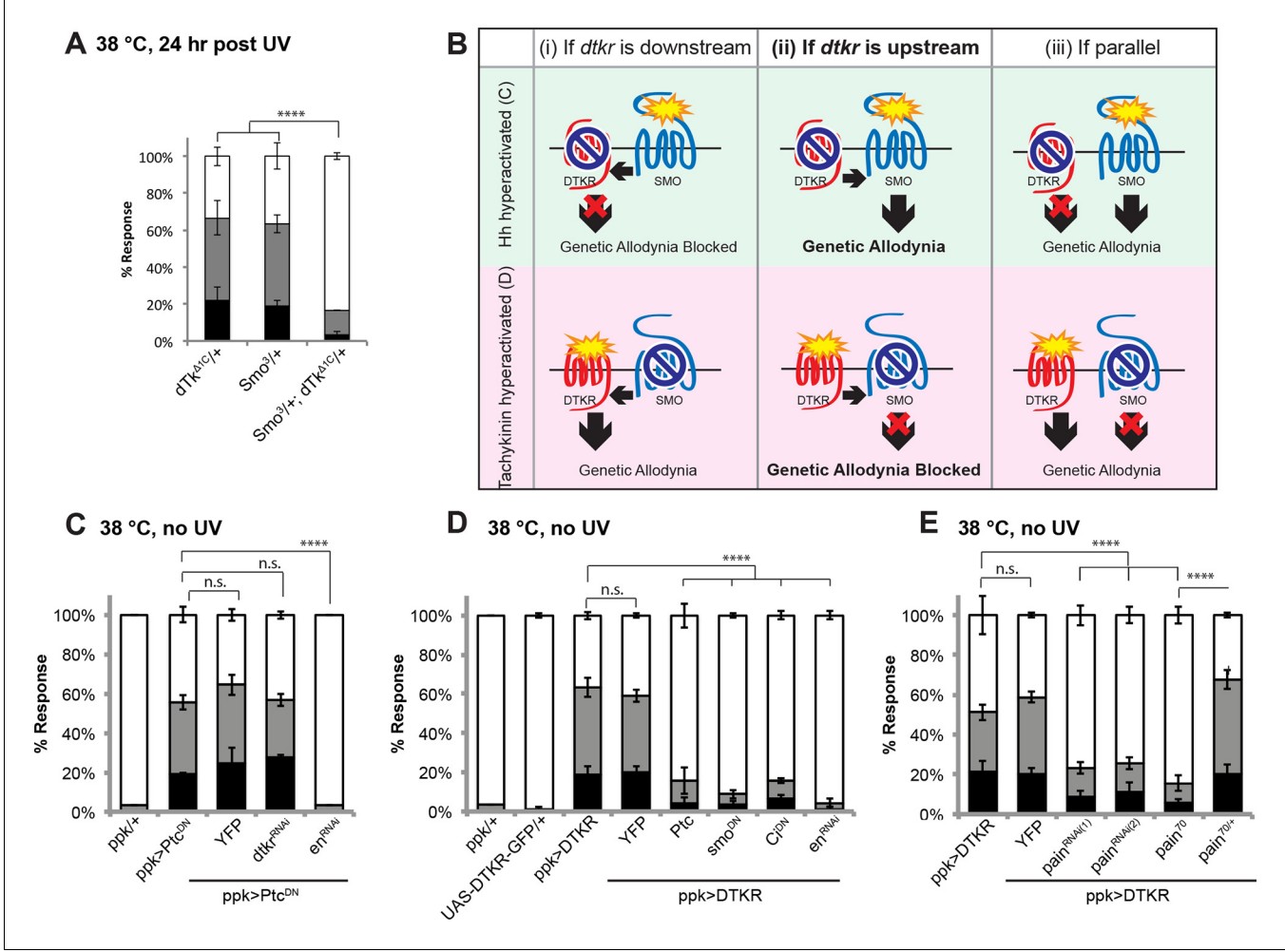

**Figure 5.** Tachykinin signaling is upstream of Smoothened and Painless in thermal allodynia. (**A**) Thermal allodynia in indicated *dTk* and *smo* heterozygotes and transheterozygotes. (**B**) Schematic of the expected results for genetic epistasis tests between the dTK and Hh pathways. (**C**) Suppression of Hh pathway-induced "genetic" allodynia by co-expression of *UAS-dtkr*[RNAi]. *UAS-en*[RNAi] serves as a positive control. (**D–E**) Suppression of DTKR-induced "genetic" allodynia. (**D**) Co-expression of indicated transgenes targeting the Hh signaling pathway and relevant controls. (**E**) Co-expression of indicated RNAi transgenes targeting TRP channel, *painless*.

The following figure supplements are available for figure 5:

**Figure supplement 1.** Alternative data presentation of thermal allodynia results (*Figure 5A* and *Figure 5D*) in non-categorical line graphs of accumulated percent response as a function of measured latency.

**Figure supplement 2.** Genetic epistasis tests between DTKR and TNF pathway.

**Figure supplement 3.** Schematic of *painless* genomic locus. *painless*[70] was generated by imprecise excision of *painless*[EP2451], deleting 4.5 kb of surrounding sequence including the ATG of the A splice variant.

**Figure supplement 4.** The *pain*[70] deletion allele and *UAS-pain*[RNAi] transgenes cause defects in baseline thermal nociception.

## Hedgehog is produced following injury in a Dispatched-dependent fashion from class IV nociceptive sensory neurons

Where does Hh itself fit into this scheme? Although *hh*[ts2] mutants show abnormal sensitization (*Babcock et al., 2011*), it remained unclear where Hh is produced during thermal allodynia. To find the source of active Hh, we tried tissue-specific knockdowns. However, none of the *UAS-Hh*[RNAi]

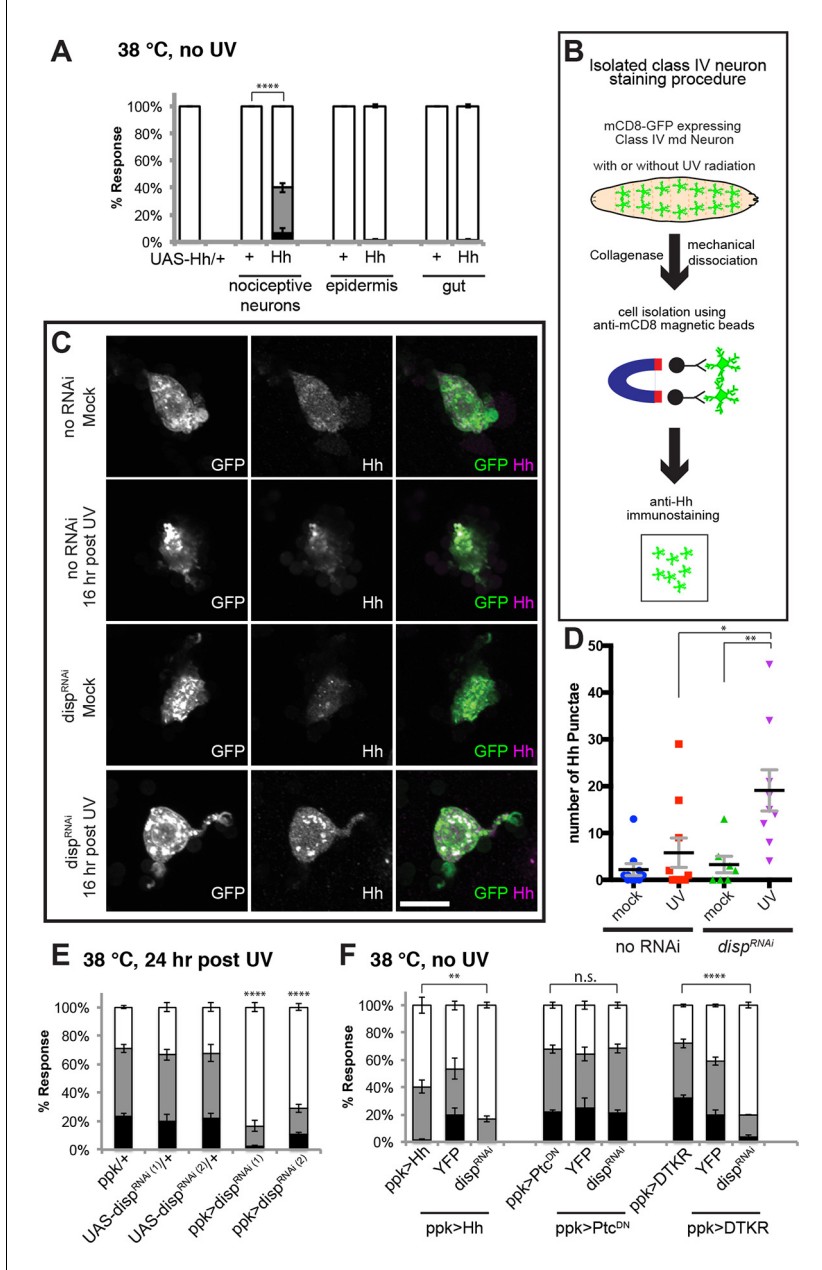

**Figure 6.** Tachykinin-induced Hedgehog is autocrine from class IV nociceptive sensory neurons. (**A**) "Genetic" allodynia induced by ectopic Hh overexpression in various tissues. Tissue-specific Gal4 drivers, UAS controls and combinations are indicated. The Gal4 drivers used are *ppk-Gal4* (class IV sensory neuron), *A58-Gal4* (epidermis), and *Myosin1A-Gal4* (gut). (**B**) Schematic of class IV neuron isolation and immunostaining. (**C**) Isolated class IV neurons stained with anti-Hh. mCD8-GFP (green in merge); anti-Hh (magenta in merge). (**D**) Number of Hh punctae in isolated class IV neurons from genotypes/conditions in (**C**). Punctae per image are plotted as individual points. Black bar; mean gray bracket; SEM. Statistical significance was determined by One-way ANOVA test followed by multiple comparisons with Tukey correction. (**E**) UV-induced thermal allodynia upon *UAS-disp^{RNAi}* expression with relevant controls. (**F**) Suppression of "genetic" allodynia by co-expression of *UAS-disp^{RNAi}* in class IV neurons. Genetic allodynia conditions were induced by Hh overexpression, Ptc^{DN} expression, or DTKR-GFP overexpression.

The following figure supplements are available for figure 6:

**Figure supplement 1.** RNAi-mediated knockdown of *hh* was not effective.

*Figure 6 continued on next page*

*Figure 6 continued*

**Figure supplement 2.** RNAi-mediated knockdown of *hh* was not effective in blocking thermal allodynia.
**Figure supplement 3.** A few more examples of isolated class IV neurons stained with anti-Hh.
**Figure supplement 4.** Genetic allodynia in the absence of tissue injury upon overexpression of TNF in class IV neurons.

transgenes we tested were effective at inducing wing patterning phenotypes in the wing imaginal disc (*Figure 6—figure supplement 1*) nor exhibited defects in thermal allodynia (*Figure 6—figure supplement 2*). Thus, we asked if tissue-specific overexpression of *UAS-Hh* in a variety of tissues could induce ectopic thermal allodynia in the absence of UV. Among class IV neurons, epidermis, and gut, overexpression of Hh only in class IV neurons resulted in ectopic sensitization (*Figure 6A*). This suggests that the class IV neurons themselves are potential Hh-producing cells.

These gain-of-function results predict that Hh might be produced in an autocrine fashion from class IV neurons following tissue injury. To monitor Hh production from class IV neurons, we performed immunostaining on isolated cells. Class IV neurons expressing mCD8-GFP were physically dissociated from intact larvae, enriched using magnetic beads conjugated with anti-mCD8 antibody, and immunostained with anti-Hh (see schematic *Figure 6B*). Mock-treated control neurons did not contain much Hh and UV irradiation increased this basal amount only incrementally (*Figure 6C* and *Figure 6—figure supplement 3*). A possible reason for this incremental increase in response to UV is that Hh is a secreted ligand. To trap Hh within class IV neurons, we asked if blocking *dispatched* (*disp*) function could trap the ligand within the neurons. Disp is necessary to process and release active cholesterol-modified Hh (*Burke et al., 1999*; *Ma et al., 2002*). Knockdown of *disp* by itself (no UV) had no effect; however combining UV irradiation and expression of *UAS-disp*$^{RNAi}$ resulted in a drastic increase in intracellular Hh punctae (*Figures 6C,D* and *Figure 6—figure supplement 3*). This suggests that class IV neurons express Hh and that blocking Dispatched function following UV irradiation traps Hh within the neuron.

Finally, we tested if trapping Hh within the class IV neurons influenced UV-induced thermal allodynia. Indeed, class IV neuron-specific expression of two non-overlapping *UAS-disp*$^{RNAi}$ transgenes each reduced UV-induced allodynia (*Figure 6E*). Furthermore, we tested whether expression of *UAS-disp*$^{RNAi}$ blocked the ectopic sensitization induced by Hh overexpression. It did (*Figure 6F*), indicating that Disp function is required for production of active Hh in class IV neurons, as in other cell types and that Disp-dependent Hh release is necessary for this genetic allodynia. *disp* function was specific; expression of *UAS-disp*$^{RNAi}$ did not block *UAS-TNF*-induced ectopic sensitization even though TNF is presumably secreted from class IV neurons in this context (*Figure 6—figure supplement 4*). Expression of *UAS-disp*$^{RNAi}$ did not block *UAS-Ptc*$^{DN}$-induced ectopic sensitization, suggesting that this does not depend on the generation/presence of active Hh (*Figure 6F*). Finally, we tested if *UAS-disp*$^{RNAi}$ expression blocked the ectopic sensitization induced by *UAS-DTKR-GFP* overexpression. It could, further supporting the idea that Disp-dependent Hh release is downstream of the Tachykinin pathway (*Figure 6F*). Thus, UV-induced tissue damage causes Hh production in class IV neurons. Dispatched function is required downstream of DTKR but not downstream of Ptc, presumably to liberate Hh ligand from the cell and generate a functional thermal allodynia response.

## Discussion

This study establishes that Tachykinin signaling regulates UV-induced thermal allodynia in *Drosophila* larvae. *Figure 7* introduces a working model for this regulation. We envision that UV radiation either directly or indirectly activates Tachykinin expression and/or release from peptidergic neuronal projections - likely those within the CNS that express DTK and are located near class IV axonal tracts. Following release, we speculate that Tachykinins diffuse to and ultimately bind DTKR on the plasma membrane of class IV neurons. This activates downstream signaling, which is mediated at least in part by a presumed heterotrimer of a G alpha (Gαq, CG17760), a G beta (Gβ5), and a G gamma (Gγ1) subunit. One likely downstream consequence of Tachykinin receptor activation is Dispatched-

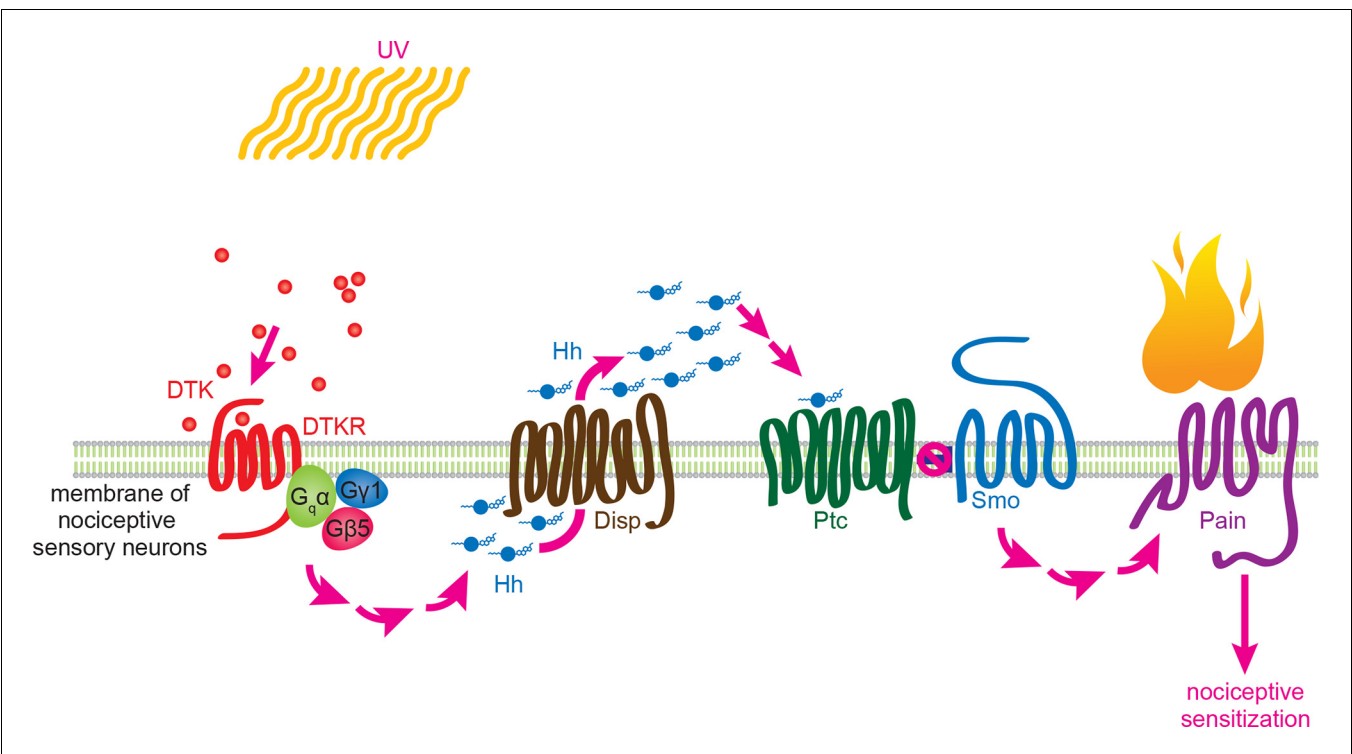

**Figure 7.** Working model for Tachykinin/Tachykinin Receptor function upstream of Hh signaling in UV-induced thermal allodynia. Tachykinin ligands are released from the brain neurons targeting class IV nociceptive sensory neurons upon UV-induced tissue damage. DTKR is coupled to trimeric G proteins and the signaling cascade then induces Disp-dependent Hh release. Hh binds to Ptc in an autocrine fashion and activates the Smo downstream signaling cascade, followed by modification/activation of Painless. These series of signaling cascades result in thermal allodynia, where stimulation at a sub-threshold temperature induces pain behaviors (thermal nociceptive sensitization).

dependent autocrine release of Hh from these neurons. We envision that Hh then binds to Patched within the same class IV neurons, leading to derepression of Smo and activation of downstream signaling through this pathway. One new aspect of the thermal allodynia response dissected here is that the transcription factors *Cubitus interruptus* and *Engrailed* act downstream of *Smo*, suggesting that, as in other Hh-responsive cells (*Briscoe and Therond, 2005*), activation of target genes is an essential component of thermal allodynia. Finally, activation of Smo impinges upon Painless through as yet undefined mechanisms to regulate thermal allodynia. Below, we discuss in more detail some of the implications of this model for Tachykinin signaling, Hh signaling, and their conserved regulation of nociceptive sensitization.

## Systemic regulation of pain sensitization by Tachykinin signaling

### Tachykinin induction and release following UV irradiation

Our results demonstrate that *Tachykinin* is required for UV-induced thermal allodynia. UV radiation may directly or indirectly trigger Tachykinin expression and/or release from the DTK-expressing neurons. Given the transparent epidermis and cuticle, direct induction mechanisms are certainly plausible. Indeed in mammals, UV radiation causes secretion of SP and CGRP from both unmyelinated c fibers and myelinated Aδ fibers nociceptive sensory afferents (*Scholzen et al., 1999*; *Seiffert and Granstein, 2002*). Furthermore, in the *Drosophila* intestine Tachykinin release is induced by nutritional and oxidative stress (*Soderberg et al., 2011*), although the effect of UV has not been examined. The exact mechanism of UV-triggered neuropeptide release remains unclear; however, we speculate that UV causes depolarization and activation of exocytosis of Tachykinin-containing vesicles.

## Ligand receptor targeting

In heterologous cells synthetic Tachykinins (DTK1-5) can activate DTKR (*Birse et al., 2006*). Our immunostaining analysis of *dTk* and genetic analysis of tissue-specific function of *dtkr* supports the model that Tachykinins from brain peptidergic neurons bind to DTKR expressed on class IV neurons. Pan-neuronal, but not class IV neuron-specific knockdown of *dTk* reduced allodynia, whereas modulation of DTKR function in class IV neurons could either decrease (RNAi) or enhance (overexpression) thermal allodynia. How do brain-derived Tachykinins reach DTKR expressed on the class IV neurons? The cell bodies and dendritic arbors of class IV neurons are located along the larval body wall (*Gao et al., 1999*; *Grueber et al., 2003*), beneath the barrier epidermal cells (*Han et al., 2014*). However, the axonal projection of each nociceptive neuron extends into the ventral nerve cord (VNC) of the CNS (*Grueber et al., 2003*; *Merritt and Whitington, 1995*) in close proximity to Tachykinin-expressing axons. Because neuropeptide transmission does not depend on specialized synaptic structures (*Zupanc, 1996*), we speculate given their proximity that Tachykinin signaling could occur via perisynaptic or volume transmission (*Agnati et al., 2006*; *Nassel, 2009*). An alternative possibility is that Tachykinins are systemically released into the circulating hemolymph (*Babcock et al., 2008*) as neurohormones (*Nassel, 2002*) following UV irradiation, either from the neuronal projections near class IV axonal tracts or from others further afield within the brain. Indeed the gain-of-function behavioral response induced by overexpression of DTKR, a receptor that has not been reported to have ligand-independent activity (*Birse et al., 2006*), suggests that class IV neurons may be constitutively exposed to a low level of subthreshold DTK peptide in the absence of injury. The direct and indirect mechanisms of DTK release are not mutually exclusive and it will be interesting to determine the relative contribution of either mechanism to sensitization.

## G protein signaling

Like most GPCRs, DTKR engages heterotrimeric G proteins to initiate downstream signaling. Gq/11 and calcium signaling are both required for acute nociception and nociceptive sensitization (*Tappe-Theodor et al., 2012*). Our survey of G protein subunits identified a putative Gαq, CG17760. Birse et al. demonstrated that DTKR activation leads to an increase in $Ca^{2+}$, strongly pointing to Gαq as a downstream signaling component (*Birse et al., 2006*). To date, *CG17760* is one of three G alpha subunits encoded in the fly genome that has no annotated function in any biological process. For the G beta and G gamma classes, we identified Gβ5 and Gγ1. Gβ5 was one of two G beta subunits with no annotated physiological function. Gγ1 regulates asymmetric cell division and gastrulation (*Izumi et al., 2004*), cell division (*Yi et al., 2006*), wound repair (*Lesch et al., 2010*), and cell spreading dynamics (*Kiger et al., 2003*). The combination of tissue-specific RNAi screening and specific biologic assays, as employed here, has allowed assignment of a function to this previously "orphan" gene in thermal nociceptive sensitization. Our findings raise a number of interesting questions about Tachykinin and GPCR signaling in general in *Drosophila*: Are these particular G protein subunits downstream of other neuropeptide receptors? Are they downstream of DTKR in biological contexts other than pain? Could RNAi screening be used this efficiently in other tissues/behaviors to identify the G protein trimers relevant to those processes?

## Hedgehog signaling as a downstream target of Tachykinin signaling

To date we have found three signaling pathways that regulate UV-induced thermal allodynia in *Drosophila* – TNF (*Babcock et al., 2009*), Hedgehog (*Babcock et al., 2011*), and Tachykinin (this study). All are required for a full thermal allodynia response to UV but genetic epistasis tests reveal that TNF and Tachykinin act in parallel or independently, as do TNF and Hh. This could suggest that in the genetic epistasis contexts, which rely on class IV neuron-specific pathway activation in the absence of tissue damage, hyperactivation of one pathway (say TNF or Tachykinin) compensates for the lack of the function normally provided by the other parallel pathway following tissue damage. While TNF is independent of Hh and DTKR, analysis of DTKR versus Hh uncovered an unexpected interdependence.

We showed that Hh signaling is downstream of DTKR in the context of thermal allodynia. Two pieces of genetic evidence support this conclusion. First, flies transheterozygous for *dTk* and *smo* displayed attenuated UV-induced thermal allodynia. Thus, the pathways interact genetically. Second, and more important for ordering the pathways, loss of canonical downstream Hh signaling

components blocked the ectopic sensitization induced by DTKR overexpression. We previously showed that loss of these same components also blocks allodynia induced by either UV or Hh hyper-activation (*Babcock et al., 2011*), suggesting that these downstream Hh components are also down-stream of DTKR. The fact that Smo is activated upon overexpression of DTKR within the same cell argues that class IV neurons may need to synthesize their own Hh following a nociceptive stimulus such as UV radiation. The data supporting an autocrine model of Hh production are three fold: (1) only class IV neuron-mediated overexpression of Hh caused thermal allodynia suggesting this tissue is fully capable of producing active Hh ligand, (2) expression of *UAS-disp^{RNAi}* within class IV neurons blocked UV- and DTKR-induced thermal allodynia, implicating a role for Disp-driven Hh secretion in these cells, and (3) the combination of *UAS-disp^{RNAi}* and UV irradiation caused accumulation of Hh punctae within class IV neurons. Disp is not canonically viewed as a downstream target of Smo and indeed, blocking *disp* did not attenuate *UAS-Ptc^{DN}*-induced or *UAS-TNF*-induced allodynia, indicat-ing that Disp is specifically required for Hh production between DTKR and Smo. Thus, Tachykinin sig-naling leads to Hh expression, Disp-mediated Hh release, or both (*Figure 7*). Autocrine release of Hh has only been demonstrated in a few non-neuronal contexts to date (*Chung and Bunz, 2013*; *Zhou et al., 2012*). This signaling architecture differs from what has been found in *Drosophila* devel-opment in two main ways. One is that DTKR is not known to play a patterning role upstream of Smo. The second is that Hh-producing cells are generally not thought to be capable of responding to Hh during the formation of developmental compartment boundaries (*Guerrero and Kornberg, 2014*; *Torroja et al., 2005*).

## What happens downstream of Smoothened activation to sensitize class IV neurons?

Ultimately, a sensitized neuron needs to exhibit firing properties that are different from those seen in the naïve or resting state. Previously, we have only examined sensitization at the behavioral level. Here we also monitored changes through extracellular electrophysiological recordings. These turned out to correspond remarkably well to behavioral sensitization. In control UV-treated larvae, nearly every temperature in the low "allodynic" range showed an increase in firing frequency in class IV neurons upon temperature ramping. *Dtkr* knockdown in class IV neurons abolished the UV-induced increase in firing frequency seen with increasing temperature and overexpression of DTKR increased the firing rate comparable to UV treatment. This latter finding provides a tidy explanation for DTKR-induced 'genetic allodynia'. The correspondence between behavior and electrophysiology argues strongly that Tachykinin directly modifies the firing properties of nociceptive sensory neurons in a manner consistent with behavioral thermal allodynia.

Genetically, knockdown of *painless* blocks *DTKR*- or *Ptc^{DN}*-induced ectopic sensitization suggest-ing that, ultimately, thermal allodynia is mediated in part via this channel. Indeed, the SP receptor Neurokinin-1 enhances TRPV1 function in primary rat sensory neurons (*Zhang et al., 2007*). Tachyki-nin/Hh activation could lead to increased Painless expression, altered Painless localization, or to post-translational modification of Painless increasing the probability of channel opening at lower temperatures. Because thermal allodynia evoked by UV and Hh-activation requires Ci and En we favor the possibility that sensitization may involve a simple increase in the expression level of Pain-less, although the above mechanisms are not mutually exclusive. Altered localization has been observed with a different TRP channel downstream of Hh stimulation; Smo activation leads to PKD2L1 recruitment to the primary cilium in fibroblasts, thus regulating local calcium dynamics of this compartment (*Delling et al., 2013*). The exact molecular mechanisms by which nociceptive sen-sitization occurs is the largest black box in the field and will take a concerted effort by many groups to precisely pin down.

## Tachykinin and substance P as regulators of nociception: what is conserved and what is not?

Our results establish that Tachykinin/SP modulation of nociception is conserved across phyla. How-ever, there are substantial differences in the architecture of this signaling axis between flies and mammals. In mammals, activation of TRP channels in the periphery leads to release of SP from the nerve termini of primary afferent C fibers in the dorsal horn (*Abbadie et al., 1997*; *Allen et al., 1997*). SP and spinal NK-1R have been reported to be required for moderate to intense baseline

nociception and inflammatory hyperalgesia although some discrepancies exist between the pharmacological and genetic knockout data (*Cao et al., 1998*; *De Felipe et al., 1998*; *Mantyh et al., 1997*; *Regoli et al., 1994*; *Woolf et al., 1998*; *Zimmer et al., 1998*). The most profound difference of *Drosophila* Tachykinin signaling anatomically is that DTK is not expressed and does not function in primary nociceptive sensory neurons. Rather, DTK is expressed in brain neurons and the larval gut (*Siviter et al., 2000*), and DTKR functions in class IV neurons to mediate thermal pain sensitization. Indeed, this raises an interesting possibility for mammalian SP studies, because nociceptive sensory neurons themselves express NK-1R (*Andoh et al., 1996*; *Brown et al., 1995*; *Segond von Banchet et al., 1999*) and SP could conceivably activate the receptor in an autocrine fashion. A testable hypothesis that emerges from our studies is that NK-1R in vertebrates might play a sensory neuron-autonomous role in regulating nociception. This possibility, while suggested by electrophysiology (*Zhang et al., 2007*) and expression studies (*Andoh et al., 1996*; *Brown et al., 1995*; *Segond von Banchet et al., 1999*) has not been adequately tested by genetic analyses in mouse to date.

In summary, we discovered a conserved role for systemic Tachykinin signaling in the modulation of nociceptive sensitization in *Drosophila*. The sophisticated genetic tools available in *Drosophila* have allowed us to uncover both a novel genetic interaction between Tachykinin and Hh signaling and an autocrine function of Hh in nociceptive sensitization. Our work thus provides a deeper understanding of how neuropeptide signaling fine-tunes an essential behavioral response, aversive withdrawal, in response to tissue damage.

## Materials and methods

### Experimental procedures

#### Fly stocks and genetics

All experimental crosses were performed at 25°C. Flies were raised on regular corn meal media. $w^{1118}$ and/or *ppk1.9-Gal4/+* (crossed to $w^{1118}$) served as control strains for behavioral analysis and staining. *dTk* mutant alleles used: $dTk^{EY21074}$, and *Df(3R)Exel7312* (*dTk* deficiency line). $dTk^{d08303}$ and $dTk^{f03824}$ insertion alleles were used to generate a custom deletion mutant of *dTk*. Detailed information regarding the generation of $dTk^{\Delta1C}$ can be found in Supplemental information. *dtkr* mutant alleles used: $dtkr^{f02797}$, $dtkr^{MB09356}$, $dtkr^{s2222}$, and *Df(3R)Exel6213* (*dtkr* deficiency line).

To make $dTk^{\Delta1C}$, a deletion allele of *dTk*, we followed FRT-mediated custom deletion methodology (*Parks et al., 2004*), using *heat-shock-Flippase*, $dTk^{d08303}$, and $dTk^{f03824}$, which are available from the Harvard Exelixis collection. We first screened deletion mutants whose eye color became stronger orange since FRT-mediated deletion resulted in a duplication of mini-white markers. Then the deletion was molecularly confirmed by PCR amplification. Primers used to confirm the deletion are listed below.

To make $painless^{70}$, a deletion allele of *painless*, we performed imprecise excision of the P element, $painless^{EP2451}$. The initial screening was based on loss of eye color pigmentation, and the deletion was molecularly confirmed by PCR and sequencing.

*Df(3R)Exel7312* was used for *dTk*, *Df(3R)Exel6213* was used for *dtkr*. The $smo^3$ (*Chen and Struhl, 1996*) allele was used to test genetic interaction with *dtk*. Tissue-specific expression of *UAS* transgenes was controlled by *ppk1.9-GAL4* for class IV md neurons (*Ainsley et al., 2003*), *A58-GAL4* for barrier epidermis (*Galko and Krasnow, 2004*), *Myosin1A-GAL4* for gut (*Jiang and Edgar, 2009*), *nubbin-GAL4* for wing imaginal disc patch (*Barrio and de Celis, 2004*), and *daughterless-GAL4* for ubiquitous expression (*Wodarz et al., 1995*). *UAS-DTKR-GFP* (*Kahsai et al., 2010b*) was used to overexpress DTKR in class IV md neurons. *UAS-smo.5A* (=*UAS-Smo$^{DN}$*) (*Collins and Cohen, 2005*), *UAS-Ptc* (*Johnson et al., 1995*), $UAS-Ci^{76}$ (=*UAS-Ci$^{DN}$*) (*Aza-Blanc et al., 1997*) were used to inhibit Hh signaling and their use in thermal nociception was previously reported (*Babcock et al., 2011*). RNAi lines used in this study are v103662 (*dTk*, RNAi (1)), v1372 (*dtkr*), v105485 (*Gsα60A*), v102887 (*CG30054*), v19124 (*Goα47A*), v105300 (*Gα49B*), v17054 (*Gα73B*), v107613 (*CG17760*), v52308 (*CG17760*), v101373 (*CG3004*), v100011 (*Gβ13F*), v104745 (*Gβ76C*), v108261 (*Gβ5*), v101733 (*RSG7*), v100140 (*Klp54D*), v102706 (*Gγ30A*), 8261R1 (*Gγ1*), v105678 (*engrailed*), v37249 (*TrpA1*), v39477 (*painless*), v10004 (*dispatched*), v42255 (*CG17760*) from Vienna RNAi center, and 25800 (*dTk*, RNAi (2)), 31132 (*concertina*), 28310 (*Gβ5*), 34372 (*Gγ1*), 31510 (*painless*), and 44633 (*dispatched*) from the Bloomington stock center.

Please see the list 'Flies used in this study' for genotypes of larvae that were used in each figure in this study

Sequence of primers used in this study
dTKdelta1C_A TACTAGGGTTAGTTCTATGGG
dTKdelta1C_B TAAACTGCGACTTGAAGCGG
dTKdelta1C_C CGTACAAATTGTGAAAGTGCC
dTKdelta1C_D TTTCAGTTGTGGTACATCTACG
dTKdelta1C_E TTGATTTAAGGTTACAGCTGTG
dTKdelta1C_F ATGCTTTGACATTTGAGAGCC
dTKdelta1C_G TGCCATTTTATCCCACCGTG
dTKdelta1C_H GTTGTTGGTTCACATTGCGTC
pain_P1 AGACGAGGAATCCAACTCGAG
pain_P2 TCGTTGATGTCTACGCGATC

## Behavioral assays

UV-induced tissue damage and thermal nociception assays were performed as described previously (*Babcock et al., 2009*; *Babcock et al., 2011*), and a brief description is the following. To induce tissue damage, early third instar larvae were etherized (Ethyl Ether Anhydrous, Fisher Scientific, Pittsburgh, PA), immobilized, and exposed to 254 nm wavelength UV at a setting of 20 mJ/cm$^2$ for about 5 sec using spectrolinker XL-1000 UV crosslinker (Spectroline, Westbury, NY). During the exposure to UV, a hand-held UV spectrophotometer (AccuMAX XS-254, Spectroline) was placed next to the specimen reading the given UV level, which usually ranges 11 – 14 mJ/cm2. Then mock or UV irradiated larvae were returned to regular fly food until thermal nociception assays were performed. The metal tip of a custom-built thermal probe, whose surface temperature is fine-tuned, touches the dorsal side of an early third instar larva in abdominal segments A4-A6. Temperature dose response curve assays were performed at a heat probe setting ranging from 38°C to 48°C with 2 degree increments. Baseline thermal nociception was assayed at heat probe settings of 45°C and 48°C in the absence of tissue damage. Thermal allodynia assays were performed at a heat probe setting of 38°C 24 hr following UV irradiation. Aversive withdrawal behavior was scored under a dissecting stereomicroscope. The corkscrew-like rolling behavior (withdrawal behavior) was monitored and the latency recorded up to a 20 s cutoff. All thermal nociception assays were performed where the experimenter was blind with respect to genotype of the animals tested. For categorical data presentation, each larva was put into one of three groups: non-responders (>20 s), slow responders ($6 \leq x \leq 20$ s), and fast responders ($\leq 5$ s). The behavioral results were tested in triplicates or more of n = 30, and tested for statistical significance using Chi-square analysis in Graphpad Prism unless noted otherwise in the figure legends. For some experiments the data was plotted non-categorically in line graphs of the accumulated percent response on the Y-axis versus latency on the X-axis, and tested for statistical significance using Log-rank (Mantel-Cox) test in Graphpad Prism.

## Electrophysiology

Extracellular recording of C4da neuronal activity was performed as described before (*Xiang et al., 2010*). UV treatment followed the same protocol as behavioral experiments. Genotypes for 3B-C: *ppk1.9-GAL4, ppk-eGFP/+*, 3D: *ppk1.9-GAL4, ppk-eGFP/+ and UAS-dtkr$^{RNAi}$/+*; *ppk1.9-GAL4, ppk-eGFP/+*, 3F: *ppk1.9-GAL4/+*, 3G: *UAS-DTKR-GFP/+; ppk1.9-GAL4/+*. 96 hr AEL third instar larvae were dissected to make fillet preparations. Fillets were prepared in external saline solution composed of (in mM): NaCl 120, KCl 3, MgCl$_2$ 4, CaCl$_2$ 1.5, NaHCO$_3$ 10, trehalose 10, glucose 10, TES 5, sucrose 10, HEPES 10. The Osmolality was 305 mOsm kg$^{-1}$ and the pH was 7.25. GFP-positive (C4da) neurons were located under a Zeiss D1 microscope with a 40X/1.0 NA water immersion objective lens. After digestion of muscles covering the C4da neurons by proteinase type XXIII (Sigma, St. Louis, MO), gentle negative pressure was applied to the C4da neuron to trap the soma in a recording pipette (5 μm tip opening; 1.5–2.0 MΩ resistance) filled with external saline solution. Recordings were performed with a 700A amplifier (Molecular Devices, Sunnyvale, CA), and the data were acquired with Digidata 1322A (Molecular Devices) and Clampex 10.5 software (Molecular Devices). Extracellular recordings of action potentials were obtained in voltage clamp mode with a holding potential of 0 mV, a 2 kHz low-pass filter and a sampling frequency of 20 kHz. For temperature

stimulation, a perfusion system delivered room temperature (RT) or pre-heated saline that flowed through the recording chamber and was removed via vacuum to maintain a constant volume. Saline was perfused at a rate of 3 mL per minute and the fillet temperature was monitored from 25–45°C using a BAT-10 electronic thermometer coupled to an IT-21 implantable probe (Physitemp, Clifton, NJ). For each recording, average firing frequency during a 3 min RT perfusion was subtracted from the average firing frequency over 1 degree bins to quantify the change in firing frequency for each temperature.

## Immunofluorescence

The primary antibodies used in this study are a guinea pig antiserum against DTK6 (a gift from David Anderson), a rabbit antiserum against the cockroach peptide LemTRP-1 (a gift from Dick Nassel), a mouse antiserum against GFP (SantaCruz, Dallas, TX), and a rabbit antiserum against Hh (a gift from Suzanne Eaton). The secondary antibodies are a Cy3-conjugated goat antiserum against guinea pig IgG (Jackson ImmunoResearch Laboratories, West Grove, PA), a Cy3-conjugated goat antiserum against rabbit IgG (Jackson ImmunoResearch Laboratories), and an Alexa488-conjugated goat antiserum against mouse IgG (Life Technologies, Grand Island, NY). Third instar larval brains and larval fillet were dissected in ice-cold PBS, fixed for one hour in 4% paraformaldehyde, and blocked for one hour in 3% normal goat serum in PBS-Tx (1X Phosphate-buffered saline with 0.3% Triton X-100). Fixed larvae were incubated overnight at 4°C in primary antibody solutions (1:1,000 dilution for anti-LemTRP-1, 1:2,000 for anti-DTK6, and 1:200 for anti-GFP in PBS-Tx), and following 5 times wash in PBS-Tx for 20 min then they were incubated overnight at 4°C in secondary antibodies solutions (1:500 dilution in PBS-Tx). After wash, stained samples were mounted in Vectashield. Images were obtained from an Olympus Fv1000 Confocal microscope. Identical settings for laser intensity and other image capture parameters were applied for comparison of Tachykinin staining in the control and mutant brains. Confocal stacks were then projected using ImageJ software, processed universally and equivalently in Photoshop.

For isolated class IV neuron immunostaining, the experimental procedure was modified from Eeger et al., (*Egger et al., 2013*) and Iyer et al., (*Iyer et al., 2009*). UV or mock treatment was as for behavioral experiments. 16 hr after UV- or mock- treatment, *ppk-Gal4>UAS-mCD8-GFP*-expressing larvae were dissected in Schneider's medium to remove gut and fat body and washed three times in 1 ml of Rinalidini solution (8 mg/ml NaCl, 0.2 mg/ml KCl, 0.05 mg/ml $NaH_2PO_4H_2O$, 1 mg/ml $NaHCO_3$, 1 mg/ml glucose, 1% pen-strep). Washed larvae were incubated in 0.5 mg/ml Collagenase I solution (Sigma) for one hour at room temperature, washed in Schneider's medium, and then mechanically dissociated by repeated pipetting. Dissociated tissue was filtered through a 40 μm cell strainer and cells were incubated with anti-mCD8a antibody-conjugated magnetic beads (eBioscience, San Diego, CA) on ice for 30 min followed by PBS washes. Isolated class IV neurons were plated on Concanavalin A (Sigma) coated coverslips and immunostained with rabbit anti-Hh antibody (1:100). Images were obtained from an Olympus Fv1000 Confocal microscope. Identical settings for laser intensity and other image capture parameters were applied for comparison of Hh staining in the control and *UAS-disp^RNAi*-expressing cells. Confocal stacks were projected using Image J, processed universally and equivalently in Photoshop and quantification was performed using the particle analysis/threshold tools in image J.

## Flies used in this study

Please note the X chromosome genotype is simplified. The actual genotypes for the X chromosome could be mixed, depending on the source RNAi collection, and the sex of individual larvae as male and female progeny were pooled together in test populations.

Figure panels – genotypes tested:
*Figure 1A* – $w^{1118}$;
*Figure 1B* – $w^{1118}$;; $dTk^{\Delta1C}$
*Figure 1C* – $w^{1118}$;; $dTk^{EY21074}$
*Figure 1D* – *elav-Gal4/+*
*elav-Gal4/+; UAS-dTk^{RNAi}* (v103662)/+
*Figure 1E* – $w^{1118}$
$w^{1118}$;; $dTk^{EY21074}$

$w^{1118}$;; $dTk^{\Delta1C}$
$w^{1118}$;; $dTk^{EY21074}$/$dTk^{\Delta1C}$
**Figure 1F** – $w^{1118}$;; ppk1.9Gal4/+
$w^{1118}$; UAS-dTk$^{RNAi}$(v103662)/+; ppk1.9Gal4/+
$w^{1118}$;; ppk1.9Gal4/UAS-dTk$^{RNAi}$ (25800)
elav-Gal4/+
$w^{1118}$; UAS-dTk$^{RNAi}$ (v103662)/+
$w^{1118}$;; UAS-dTk$^{RNAi}$ (25800)/+
elav-Gal4/+; UAS-dTk$^{RNAi}$ (v103662)/+
elav-Gal4/+;; UAS-dTk$^{RNAi}$ (25800)/+
**Figure 1G** – $w^{1118}$
$w^{1118}$;; $dTk^{EY21074}$/+
$w^{1118}$;; $dTk^{\Delta1C}$/+
$w^{1118}$;; $dTk^{EY21074}$
$w^{1118}$;; $dTk^{\Delta1C}$
$w^{1118}$;; $dTk^{EY21074}$/$dTk^{\Delta1C}$
$w^{1118}$;; Df(3R)Exel7312/+
$w^{1118}$;; $dTk^{EY21074}$/Df(3R)Exel7312
$w^{1118}$;; $dTk^{\Delta1C}$/Df(3R)Exel7312
**Figure 1** FS1– $w^{1118}$; ppk-eGFP/+
**Figure 1** FS2– elav-Gal4/+
elav-Gal4/+; UAS-dTk$^{RNAi}$ (v103662)/+
**Figure 1** FS3 – $w^{1118}$
$w^{1118}$;; $dTk^{\Delta1C}$
**Figure 1** FS4– $w^{1118}$
**Figure 1** FS5– elav-Gal4/+
$w^{1118}$; UAS-dTk$^{RNAi}$ (v103662)/+
$w^{1118}$;; UAS-dTk$^{RNAi}$ (25800)/+
elav-Gal4/+; UAS-dTk$^{RNAi}$ (v103662)/+
elav-Gal4/+;; UAS-dTk$^{RNAi}$ (25800)/+
$w^{1118}$
$w^{1118}$;; $dTk^{EY21074}$/+
$w^{1118}$;; $dTk^{\Delta1C}$/+
$w^{1118}$;; $dTk^{EY21074}$/Df(3R)Exel7312
$w^{1118}$;; $dTk^{\Delta1C}$/Df(3R)Exel7312
**Figure 2B** – $w^{1118}$;; ppk1.9Gal4/+
$w^{1118}$;; ppk1.9Gal4/UAS-dtkr$^{RNAi}$(v1372)
**Figure 2C** – $w^{1118}$
$w^{1118}$;; dtkr$^{f02797}$
$w^{1118}$;; dtkr$^{MB09356}$
**Figure 2D** – $w^{1118}$;; ppk1.9Gal4/+
$w^{1118}$;; UAS-dtkr$^{RNAi}$(v1372)/+
$w^{1118}$;; ppk1.9Gal4/UAS-dtkr$^{RNAi}$(v1372)
$w^{1118}$; UAS-DTKR-GFP/+; ppk1.9Gal4/UAS-dtkr$^{RNAi}$(v1372)
**Figure 2E** – $w^{1118}$
$w^{1118}$;; dtkr$^{f02797}$/+
$w^{1118}$;; dtkr$^{MB09356}$/+
$w^{1118}$;; dtkr$^{s2222}$/+
$w^{1118}$;; Df(3R)Exel6213/+
$w^{1118}$;; dtkr$^{f02797}$
$w^{1118}$;; dtkr$^{MB09356}$
$w^{1118}$;; dtkr$^{s2222}$/dtkr$^{MB09356}$
$w^{1118}$;; dtkr$^{f02797}$/Df(3R)Exel6213
$w^{1118}$;; dtkr$^{MB09356}$/Df(3R)Exel6213
$w^{1118}$;; dtkr$^{s2222}$/Df(3R)Exel6213
$w^{1118}$;; dtkr$^{f02797}$/dtkr$^{MB09356}$

$w^{1118}$;; ppk1.9Gal4, dtkr$^{f02797}$/dtkr$^{MB09356}$
$w^{1118}$; UAS-DTKR-GFP/+; dtkr$^{f02797}$/dtkr$^{MB09356}$
$w^{1118}$; UAS-DTKR-GFP/+; ppk1.9Gal4, dtkr$^{f02797}$/dtkr$^{MB09356}$
*Figure 2F – $w^{1118}$;; ppk1.9Gal4/+*
$w^{1118}$; UAS-DTKR-GFP/+
$w^{1118}$; UAS-DTKR-GFP/+; ppk1.9Gal4/+
*Figure 2G~I – $w^{1118}$; UAS-DTKR-GFP/+; ppk1.9Gal4/+*
*Figure 2 FS1– $w^{1118}$;; ppk1.9Gal4/+*
$w^{1118}$;; UAS-dtkr$^{RNAi}$(v1372)/+
$w^{1118}$;; ppk1.9Gal4/UAS-dtkr$^{RNAi}$(v1372)
$w^{1118}$; UAS-DTKR-GFP/+; ppk1.9Gal4/UAS-dtkr$^{RNAi}$(v1372)
$w^{1118}$
$w^{1118}$;; dtkr$^{f02797}$/dtkr$^{MB09356}$
$w^{1118}$;; ppk1.9Gal4, dtkr$^{f02797}$/dtkr$^{MB09356}$
$w^{1118}$; UAS-DTKR-GFP/+; dtkr$^{f02797}$/dtkr$^{MB09356}$
$w^{1118}$; UAS-DTKR-GFP/+; ppk1.9Gal4, dtkr$^{f02797}$/dtkr$^{MB09356}$
*Figure 3B~D – $w^{1118}$;; ppk1.9Gal4, ppk-eGFP/+*
*Figure 3E – $w^{1118}$;; ppk1.9Gal4, ppk-eGFP/UAS-dtkr$^{RNAi}$(v1372)*
$w^{1118}$;; dtkr$^{f02797}$/dtkr$^{MB09356}$
$w^{1118}$; UAS-DTKR-GFP/+; ppk1.9Gal4, dtkr$^{f02797}$/dtkr$^{MB09356}$
*Figure 3F~H – $w^{1118}$;; ppk1.9Gal4/+*
$w^{1118}$; UAS-DTKR-GFP/+; ppk1.9Gal4/+
Figure 3 FS1 – $w^{1118}$;; dtkr$^{f02797}$/+
$w^{1118}$;; ppk1.9Gal4, dtkr$^{f02797}$/dtkr$^{MB09356}$
*Figure 4B – $w^{1118}$;; ppk1.9Gal4/+*
$w^{1118}$; UAS-RNAi/+; ppk1.9Gal4/ + (if RNAi is on the second) or
$w^{1118}$;; ppk1.9Gal4/UAS-RNAi (if RNAi is on the third)
*Figure 4C – $w^{1118}$;; ppk1.9Gal4/+*
$w^{1118}$; UAS-CG17760$^{RNAi}$ (v107613)/+; ppk1.9Gal4/+
$w^{1118}$;; ppk1.9Gal4/UAS-CG17760$^{RNAi}$ (v52308)
$w^{1118}$; UAS-Gbeta5$^{RNAi}$ (v108261)/+; ppk1.9Gal4/+
$w^{1118}$;; ppk1.9Gal4/UAS-Gbeta5$^{RNAi}$ (28310)
$w^{1118}$;; ppk1.9Gal4/UAS-Ggamma1$^{RNAi}$ (8261R-1)
$w^{1118}$;; ppk1.9Gal4/UAS-Ggamma1$^{RNAi}$ (34372)
*Figure 4D – $w^{1118}$;; ppk1.9Gal4/+*
$w^{1118}$; UAS-YFP/+; ppk1.9Gal4/+
$w^{1118}$; UAS-DTKR-GFP/UAS-CG17760$^{RNAi}$ (v107613); ppk1.9Gal4/+
$w^{1118}$; UAS-DTKR-GFP/+; ppk1.9Gal4/UAS-CG17760$^{RNAi}$ (v52308)
$w^{1118}$; UAS-DTKR-GFP/UAS-Gbeta5$^{RNAi}$ (v108261); ppk1.9Gal4/+
$w^{1118}$; UAS-DTKR-GFP/+; ppk1.9Gal4/UAS-Gbeta5$^{RNAi}$ (28310)
$w^{1118}$; UAS-DTKR-GFP/+; ppk1.9Gal4/UAS-Ggamma1$^{RNAi}$ (8261R-1)
$w^{1118}$; UAS-DTKR-GFP/+; ppk1.9Gal4/UAS-Ggamma1$^{RNAi}$ (34372)
*Figure 4 FS1– $w^{1118}$;; ppk1.9Gal4/+*
$w^{1118}$; UAS-RNAi/+; ppk1.9Gal4/ + (if RNAi is on the second) or
$w^{1118}$;; ppk1.9Gal4/UAS-RNAi (if RNAi is on the third)
*Figure 4 FS2 – $w^{1118}$; UAS-CG17760$^{RNAi}$ (v107613)/+*
$w^{1118}$;; UAS-CG17760$^{RNAi}$ (v52308)/+
$w^{1118}$; UAS-Gbeta5$^{RNAi}$ (v108261)/+
$w^{1118}$;; UAS-Gbeta5$^{RNAi}$ (28310)/+
$w^{1118}$;; UAS-Ggamma1$^{RNAi}$ (8261R-1)/+
$w^{1118}$;; UAS-Ggamma1$^{RNAi}$ (34372)/+
*Figure 5A – $w^{1118}$;; dTk$^{\Delta1C}$/+*
w; smo$^3$ b$^1$ pr$^1$/+
w; smo$^3$ b$^1$ pr$^1$/+; dTk$^{\Delta1C}$/+
*Figure 5C – $w^{1118}$;; ppk1.9Gal4/+*
$w^{1118}$; UAS-Ptc$^{1130x}$/+; ppk1.9Gal4/+

$w^{1118}$; UAS-Ptc$^{1130x}$/UAS-YFP; ppk1.9Gal4/+
$w^{1118}$; UAS-Ptc$^{1130x}$/+; ppk1.9Gal4/UAS-dtkr$^{RNAi}$(v1372)
$w^{1118}$; UAS-Ptc$^{1130x}$/UAS-en$^{RNAi}$(v105678); ppk1.9Gal4/+
**Figure 5D** – $w^{1118}$;; ppk1.9Gal4/+
$w^{1118}$; UAS-DTKR-GFP/+
$w^{1118}$; UAS-DTKR-GFP/+; ppk1.9Gal4/+
$w^{1118}$; UAS-DTKR-GFP/UAS-YFP; ppk1.9Gal4/+
$w^{1118}$; UAS-DTKR-GFP/+; ppk1.9Gal4/UAS-Ptc
$w^{1118}$; UAS-DTKR-GFP/UAS-smo.5A; ppk1.9Gal4/+
$w^{1118}$; UAS-DTKR-GFP/+; ppk1.9Gal4/UAS-Ci$^{76}$
$w^{1118}$; UAS- DTKR-GFP/UAS-en$^{RNAi}$(v105678); ppk1.9Gal4/+
**Figure 5E** – $w^{1118}$; UAS-DTKR-GFP/+; ppk1.9Gal4/+
$w^{1118}$; UAS-DTKR-GFP/UAS-YFP; ppk1.9Gal4/+
$w^{1118}$; UAS-DTKR-GFP/+; ppk1.9Gal4/UAS-pain$^{RNAi}$(v39477)
$w^{1118}$; UAS-DTKR-GFP/+; ppk1.9Gal4/UAS-pain$^{RNAi}$(31510)
$w^{1118}$; UAS-DTKR-GFP, pain$^{70}$/pain$^{70}$; ppk1.9Gal4/+
$w^{1118}$; UAS-DTKR-GFP, pain$^{70}$/+; ppk1.9Gal4, /+
**Figure 5 FS1**– $w^{1118}$;; dTk$^{Δ1C}$/+
w; smo$^3$ b$^1$ pr$^1$/+
w; smo$^3$ b$^1$ pr$^1$/+; dTk$^{Δ1C}$/+
$w^{1118}$; UAS-DTKR-GFP/+; ppk1.9Gal4/+
$w^{1118}$; UAS-DTKR-GFP/+; ppk1.9Gal4/UAS-Ptc
$w^{1118}$; UAS-DTKR-GFP/UAS-smo.5A; ppk1.9Gal4/+
$w^{1118}$; UAS-DTKR-GFP/+; ppk1.9Gal4/UAS-Ci$^{76}$
$w^{1118}$; UAS- DTKR-GFP/UAS-en$^{RNAi}$(v105678); ppk1.9Gal4/+
**Figure 5 FS2**– $w^{1118}$; egr$^{Regg1c}$/+; ppk1.9Gal4/+
$w^{1118}$; egr$^{Regg1c}$/+; ppk1.9Gal4/ UAS-dtkr$^{RNAi}$(v1372)
$w^{1118}$; UAS-DTKR-GFP/+; ppk1.9Gal4/+
$w^{1118}$; UAS-DTKR-GFP/UAS-wgn$^{RNAi}$; ppk1.9Gal4/+
**Figure 5 FS3**– $w^{1118}$
$w^{1118}$; pain$^{70}$
**Figure 5 FS4**– $w^{1118}$
$w^{1118}$; pain$^{70}$
$w^{1118}$;; ppk1.9Gal4/+
$w^{1118}$;; UAS-pain$^{RNAi}$(v39477)/+
$w^{1118}$;; UAS-pain$^{RNAi}$(31510)/+
$w^{1118}$;; ppk1.9Gal4/UAS-pain$^{RNAi}$(v39477)
$w^{1118}$;; ppk1.9Gal4/UAS-pain$^{RNAi}$(31510)
**Figure 6A** – $w^{1118}$; UAS-Hh/+
$w^{1118}$;; ppk1.9Gal4/+
$w^{1118}$; UAS-Hh/+; ppk1.9Gal4/+
$w^{1118}$;; A58-Gal4/+
$w^{1118}$; UAS-Hh/+; A58-Gal4/+
$w^{1118}$; Myosin1A-Gal4/+
$w^{1118}$; Myosin1A-Gal4/UAS-Hh
**Figure 6C** – $w^{1118}$;; ppk1.9Gal4, UAS-mCD8-GFP/+
$w^{1118}$; UAS-disp$^{RNAi}$(v10004)/+; ppk1.9Gal4, UAS-mCD8-GFP/+
**Figure 6E** – $w^{1118}$;; ppk1.9Gal4/+
$w^{1118}$; UAS-disp$^{RNAi}$(v10004)/+
$w^{1118}$;; UAS-disp$^{RNAi}$(44633)/+
$w^{1118}$; UAS-disp$^{RNAi}$(v10004)/+; ppk1.9Gal4/+
$w^{1118}$;; ppk1.9Gal4/UAS-disp$^{RNAi}$(44633)
**Figure 6F** – $w^{1118}$; UAS-Hh/+; ppk1.9Gal4/+
$w^{1118}$; UAS-Hh/UAS-YFP; ppk1.9Gal4/+
$w^{1118}$; UAS-Hh/UAS-disp$^{RNAi}$(v10004); ppk1.9Gal4/+
$w^{1118}$; UAS-Ptc$^{1130x}$/+; ppk1.9Gal4/+

$w^{1118}$; UAS-Ptc$^{1130x}$/UAS-YFP; ppk1.9Gal4/+
$w^{1118}$; UAS-Ptc$^{1130x}$/UAS-disp$^{RNAi}$(v10004); ppk1.9Gal4/+
$w^{1118}$; UAS-DTKR-GFP/+; ppk1.9Gal4/+
$w^{1118}$; UAS-DTKR-GFP/UAS-YFP; ppk1.9Gal4/+
$w^{1118}$; UAS-DTKR-GFP/UAS-disp$^{RNAi}$(v10004); ppk1.9Gal4/+
*Figure 6* FS1 – $w^{1118}$; nubbin-Gal4/+
$w^{1118}$; nubbin-Gal4/+; UAS-Ci$^{76}$/+
$w^{1118}$; nubbin-Gal4/+; UAS-hh$^{RNAi}$/+ (v1402)
$w^{1118}$; nubbin-Gal4/+; UAS-hh$^{RNAi}$/+ (v1403)
$w^{1118}$; nubbin-Gal4/+; UAS-hh$^{RNAi}$/+ (31042)
$w^{1118}$; nubbin-Gal4/+; UAS-hh$^{RNAi}$/+ (31475)
$w^{1118}$; nubbin-Gal4/+; UAS-hh$^{RNAi}$/+ (25794)
$w^{1118}$; nubbin-Gal4/+; UAS-hh$^{RNAi}$/+ (4637R-2)
$w^{1118}$;; da-Gal4/UAS-hh$^{RNAi}$(4637R-2)
*Figure 6* FS2 – $w^{1118}$;; ppk1.9Gal4/+
$w^{1118}$;; ppk1.9Gal4/UAS-hh$^{RNAi(1)}$(4637R-2)
$w^{1118}$;; ppk1.9Gal4/UAS-hh$^{RNAi(2)}$(v1403)
$w^{1118}$;; ppk1.9Gal4/UAS-hh$^{RNAi(1+2)}$(4637R-2 + v1403)
*Figure 6* FS3– $w^{1118}$;; ppk1.9Gal4, UAS-mCD8-GFP/+
$w^{1118}$; UAS-disp$^{RNAi}$(v10004)/+; ppk1.9Gal4, UAS-mCD8-GFP/+
*Figure 6* FS4 – $w^{1118}$; egr$^{Regg1c}$/+; ppk1.9Gal4/+
$w^{1118}$; egr$^{Regg1c}$/UAS-disp$^{RNAi}$(v10004); ppk1.9Gal4/+

## Acknowledgements

We thank members of the Gutstein and Galko laboratories for comments on the manuscript, David Anderson, Andreas Bergmann, Ryan Birse, Suzanne Eaton, Georg Halder, Dick Nässel, Matt Scott, Michael Welsh for fly stocks and antibodies, the Bloomington, Harvard, NIG-fly, VDRC Stock Centers for *Drosophila* stocks, and Leisa McCord for graphics assistance. Grants that supported this study are NIH T32 CA9299-33 (SHI), NIH T32-HD07325-16 (DTB), NIH R01 NS069828 (MJG), NIH R21 NS087360 (MJG), NIH R01 NS089787 (YX), and RGY0090/2014 Human Frontier Science Program (YX). The authors declare no conflict of interest.

## Additional information

### Funding

| Funder | Grant reference number | Author |
|---|---|---|
| National Institute of Neurological Disorders and Stroke | R01 NS069828 | Michael J Galko |
| Human Frontier Science Program | RGY0090/2014 | Yang Xiang |
| National Institute of Neurological Disorders and Stroke | R01 NS089787 | Yang Xiang |
| National Institute of Neurological Disorders and Stroke | R21 NS087360 | Michael J Galko |

The funders had no role in study design, data collection and interpretation, or the decision to submit the work for publication.

### Author contributions

SHI, Conception and design, Acquisition of data, Analysis and interpretation of data, Drafting or revising the article; KT, JJ, Acquisition of data, Analysis and interpretation of data, Drafting or revising the article; DTB, Conception and design, Acquisition of data; ZM, Conception and design,

Contributed unpublished essential data or reagents; YX, MJG, Conception and design, Analysis and interpretation of data, Drafting or revising the article, Obtained funding

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
