## [Decision Letter]

Thank you for submitting your work entitled "Tachykinin acts upstream of autocrine Hedgehog signaling during nociceptive sensitization in *Drosophila*" for peer review at *eLife*. Your submission has been favorably evaluated by K VijayRaghavan (Senior editor), Mani Ramaswami (Reviewing editor), and two reviewers.

The reviewers have discussed the reviews with one another and the Reviewing editor has drafted this decision to help you prepare a revised submission.

This is a very nice paper on the mechanism of pain sensitization in *Drosophila*. The manuscript by Im et al. investigates the molecular pathways leading from UV damage to allodynia in the *Drosophila* larvae. Previous work from this group has implicated the gene hedgehog as being necessary for the induction of allodynia. The current manuscript greatly extends this model. Based on their observations, the authors concluded that upon UV treatment, *dTk* is released from brain neurons, activates Hedgehog (Hh) signaling in class IV neurons through *dTk* receptor (DTKR), and leads to Painless-dependent nociceptive sensitization. DTKR activates G-coupled protein receptors and ultimately leads to the production of hedgehog and the dis-inhibition of *smo*. Smo regulates the gene, pain, and results in increased class IV nociceptive neuron activation and behavioral allodynia. The evidence for this proposed model comes from behavioral assays using RNAi transgenes, genetic mutants, and electrophysiology. The experiments are logical, clearly explained, and proper controls are generally used. The figures are well displayed and we particularly appreciated the easy-to-follow schematic of their working model in Figure 7.

Overall, this is a carefully designed and conducted study (e.g. analysis of relevant controls and multiple mutant alleles). The results are clear and the findings are of considerable interest. I do not have serious concerns regarding methodology, experimental results and their interpretation. For the benefit of readers, however, some points need clarification and further justification. These are described below.

Concerns that must be addressed:

1) The current methodology has the experimenter press a probe on the larvae to heat the animal. How can the amount of force applied by the probe be controlled prior to or during heat application? Would this effect the latency to rolling in allodynia? This methodology appears to be fairly common in the field of pain research, but it can likely be improved moving forward. Finally, if the behavioral and electrophysiological experiments were conducted in a blinded manner, the authors need to explicitly state this in the Materials and methods. If the experiments were not blinded, then is there reason to worry that this combined with the intimate interaction between researcher and larvae and the categorical nature of the statistics could lead to large unintentional biases in data?

2) As shown in Figure 6—figure supplement 3, expression of *UAS-disp^RNAi^* does not block *UAS-TNF*-induced ectopic sensitization. Based on this observation, one would suspect that suppression of the DTK/DTKR/Hh pathway (equivalent to expression of *UAS-disp^RNAi^*) alone does not block UV-induced allodynia, because the TNF-induced pathway is activated after UV irradiation. However, this study demonstrates that suppression of the DTK/DTKR/Hh pathway (e.g. loss-of-function of *dTk* or *dtkr*) blocks UV-induced allodynia even in the intact TNF-induced pathway. The authors should provide a comprehensive picture, by further discussing and justifying the relationship between two potentially parallel pathways (i.e. DTK/DTKR/Hh and TNF pathways) that are involved in UV-induced nociceptive sensitization.

3) A broad concern that the authors should think about and consider regards the quantification of the behavioral data, even if some of these analyses are typical in the field. It appears possible that categorizing the latency data into arbitrary groups of slow, fast, or non-responders could create large statistical differences depending on the latency cut-offs that might not exist if the raw latencies were examined. Ideally, one would simply report the latencies of each animal and perform their statistics to this data. (Statistical methods for dealing with animals that did not respond do exist). Even better would be an assay that requires far less interaction between the experimenter and the larvae to prevent potential sources of accidental bias.

---

## [Author Response]

*Concerns that must be addressed:*

*1) The current methodology has the experimenter press a probe on the larvae to heat the animal. How can the amount of force applied by the probe be controlled prior to or during heat application? Would this effect the latency to rolling in allodynia? This methodology appears to be fairly common in the field of pain research, but it can likely be improved moving forward. Finally, if the behavioral and electrophysiological experiments were conducted in a blinded manner, the authors need to explicitly state this in the Methods. If the experiments were not blinded, then is there reason to worry that this combined with the intimate interaction between researcher and larvae and the categorical nature of the statistics could lead to large unintentional biases in data?*

The reviewers are correct on two fronts: that assays involving probes or plates that contact the test animal are currently standard in the field (Im and Galko, 2012; Wilson and Mogil, 2001) and that they are certainly improvable moving forward. That said, our group has already made a number of improvements to the original heat probe assay published by Tracey et al. (Babcock et al., 2009; Chattopadhyay et al., 2012; Tracey et al., 2003). The major one is the introduction of feedback control to the probe so that the setpoint temperature remains essentially constant throughout application. A minor one relevant here is the use of machine-rounded tips to minimize any sharpness that might provoke mechanical responses. One can imagine potentially more optimal ways of delivering heat (perhaps a focused laser?) (Ohyama et al., 2013) but these would need to be built in such a way (for precise sensitization experiments) that one could control the larval surface temperature predictably (and hold it constant) and so that the experimenter could track the laser on a moving larva. Such a device would eliminate the variable of touch/pressure that currently exists.

Can the amount of force applied be controlled? While we do not measure the applied force directly in our assay we can provide the following reassurances. First, the applied force is not sufficient to provoke any noxious mechanical aversive response – which is the same rolling that is observed with heat (Babcock et al., 2009; Chattopadhyay et al., 2012). Probes set to 38 ºC and below (sub-noxious temperatures) do not provoke aversive rolling. Second, in our lab multiple independent probe users produce temperature versus behavior dose-response curves that are (within the constraints of a behavioral assay) remarkably similar – this data is now provided in Figure 1—figure supplement 4. These curves are characterized by no responsiveness below 39 ºC, nearly 100% fast responders at 48ºC, and graded responses (the exact grades can vary some) of fast and slow responders in between these limits. These results suggest that for practiced probe users the sub-noxious force applied is not a major contributor to the behavioral output even if the applied sub-noxious force varies. The main major contributor is the intensity of the input temperature.

A second question posed is whether variations in the applied force might affect the latency of the allodynia response? This is conceivable but somewhat unlikely. Because in our assay the applied force is sub-threshold and is not sufficient to provoke aversive behavior we doubt that it is making a major contribution to the observed latency responses. Indeed, the fact that the temperature versus aversive behavior dose-response data tracks remarkably tightly with temperature suggests that any variations in force of probe application can have a minimal effect on the behavioral output, at most.

Finally, the reviewers ask us to address blinding. All behavioral experiments were performed where the investigator was blind with respect to the genotype of the larvae being tested and this is now noted in the methods section. Because of this methodology unintentional expectational bias would be difficult to arise. We address the categorical statistics further below.

*2) As shown in Figure 6—figure supplement 3, expression of* UAS-disp^RNAi^
*does not block* UAS-TNF*-induced ectopic sensitization. Based on this observation, one would suspect that suppression of the DTK/DTKR/Hh pathway (equivalent to expression of* UAS-disp^RNAi^*) alone does not block UV-induced allodynia, because the TNF-induced pathway is activated after UV irradiation. However, this study demonstrates that suppression of the DTK/DTKR/Hh pathway (e.g. loss-of-function of* dTk *or* dtkr*) blocks UV-induced allodynia even in the intact TNF-induced pathway. The authors should provide a comprehensive picture, by further discussing and justifying the relationship between two potentially parallel pathways (i.e. DTK/DTKR/Hh and TNF pathways) that are involved in UV-induced nociceptive sensitization.*

The data provided in Figure 6—figure supplement 3 (now in Figure 6—figure supplement 4) was meant to reassure that Dispatched is specific for Hh secretion from class IV neurons – that loss of Dispatched does not block TNF secretion from these cells when TNF is ectopically expressed there. But we certainly see how this could lead to some confusion. Perhaps the best way to deal with the potential confusion is to add more data that at least clarifies the relationship between TNF and Tachykinin/Hh signaling. Our previous study established that even though TNF and Hh signaling are both required for UV-induced thermal allodynia, and even though they act in parallel in terms of genetic epistasis (similar analysis to that performed here where each pathway is activated and the other blocked) blocking either pathway alone is sufficient to curtail UV-induced thermal allodynia (Babcock et al., 2011). The relationship between TNF and Tachykinin signaling is similar, perhaps not surprisingly since Tachykinin is upstream of Hh. We have added data (Figure 5—figure supplement 2) to show that knocking down TNFR (*UAS-wengen^RNAi^*) is not sufficient to block DTKR-induced genetic thermal allodynia and knocking down *dtkr (UAS-dtkr^RNAi^*) is not sufficient to block TNF signaling-induced genetic thermal allodynia. We have added in the Discussion a brief consideration of how these various epistasis results might come about.

*3) A broad concern that the authors should think about and consider regards the quantification of the behavioral data, even if some of these analyses are typical in the field. It appears possible that categorizing the latency data into arbitrary groups of slow, fast, or non-responders could create large statistical differences depending on the latency cut-offs that might not exist if the raw latencies were examined. Ideally, one would simply report the latencies of each animal and perform their statistics to this data. (Statistical methods for dealing with animals that did not respond do exist). Even better would be an assay that requires far less interaction between the experimenter and the larvae to prevent potential sources of accidental bias.*

Admittedly, the boundaries of the chosen categories (20 s cutoff for nonresponders, 5 s boundary between fast and slow) are somewhat arbitrary. But not entirely. These were chosen when we were originally preparing our first manuscript describing sensitization in *Drosophila* (Babcock et al., 2009). At the high end we felt that the standard 10 s cut off discarded a fairly large number of responders, especially in the lower noxious temperature ranges. The boundaries of the fast/slow categories were admittedly chosen to generate the most attractive dose-response curve with control larvae. All boundary options work (especially those between 4 and 8 s) – it just determines whether the dose-response curve is skewed towards the slow/non-noxious side or the fast/noxious side. Five seconds is aesthetically well-centered. The reviewers are correct that there are statistical ways to treat the non-responder class and we are thankful for prompting us to more thoroughly investigate these. Here, we present a way of doing this that converts the raw latency data into a curve that is similar conceptually to an inverted “survival” or “lifespan” curve. This has the virtue of capturing the precise latency information for every larva tested. At a number of points throughout the manuscript we have converted some of the categorical data to this form (Figure 1—figure supplement 5; Figure 2—figure supplement 1; Figure 4—figure supplement 1; Figure 5—figure supplement 1; Figure 5—figure supplement 2). In only one case did the enhanced statistical power of comparing these curves (Mantel-Cox Test) reveal a difference that was obscured by the categorical data. This was the case of *G-protein β5*, where our initial pilot screen (Figure 4) analyzed categorically showed this gene to be on the margin of statistical significance. When this data was re-analyzed non-categorically, *Gβ5* was clearly significant even at n = 30, highlighting the power of capturing all the latency data for the statistical comparison, as the reviewers anticipated.

References

Wilson, S. G. & Mogil, J. S. Measuring pain in the (knockout) mouse: big challenges in a small mammal. Behav. Brain Res. 125, 65-73 (2001).

Chattopadhyay, A., Gilstrap, A. V. & Galko, M. J. Local and global methods of assessing thermal nociception in *Drosophila* larvae. J Vis Exp, doi:10.3791/3837 (2012).